# Exploring the impact of visual function degradation on manual prehension movements in normal-sighted individuals

Pablo Sanz Diez[1,2*◦], Sandra Gisbert[1,2*◦], Annalisa Bosco[3,4], Augusto Arias[2], Patrizia Fattori[3,4], Siegfried Wahl[1,2*]

**1** Carl Zeiss Vision International GmbH, Aalen, Germany, **2** Institute for Ophthalmic Research, Eberhard Karls University Tübingen, Tübingen, Germany, **3** Department of Biomedical and Neuromotor Sciences, University of Bologna, Bologna, Italy, **4** Alma Mater Research Institute for Human-Centered Artificial Intelligence (Alma Human AI), University of Bologna, Bologna, Italy

◦ These authors contributed equally to this work.
* pablo.sanzdiez@zeiss.com (PSD); sandra.gisbert@zeiss.com (SG); siegfried.wahl@uni-tuebingen.de (SW)

## Abstract

Impairments of visual function abilities, such as visual acuity and contrast sensitivity, can negatively impact our ability to perform manual prehension tasks. Despite the clear link between visual input and motor output, there is still limited understanding of how visual function deficits affect hand motor behavior. This study aimed to explore the impact of different levels of visual function degradation, specifically in terms of visual acuity and contrast sensitivity, on the reach and grasp components of manual prehension. To this end, visual function degradation was induced in young participants with normal vision using five different densities of Bangerter occlusion foils. Participants were instructed to perform a natural and accurate reach-to-grasp task towards a cylindrical object with two different diameters (3.5 or 7 cm) and positioned at two distances (25 or 50 cm). The effects of visual function degradation, object size, and distance were evaluated by recording the position and trajectory of the right hand using an optoelectronic motion capture system. Three-dimensional kinematic analysis revealed that visual function degradation in normal-sighted individuals directly altered the reach and grasp components of prehension movements. These alterations included longer movement durations, lower velocity and acceleration profiles, slower deceleration phases, over-scaled hand grip apertures, and greater trajectory deviations. The effects were dependent on the level of visual degradation induced and the intrinsic (size) and extrinsic (distance) object properties. Reductions exceeding 70% in visual acuity and 55% in CS had the most pronounced impact on prehension components. However, subtle reductions greater than 30% in visual acuity and 15% in contrast sensitivity were sufficient to trigger compensatory mechanisms. These findings provide further understanding of how visual function degradation affects

**Data availability statement:** All data are available from the Open Science Framework database (https://osf.io/wrja5/).

**Funding:** This work was supported by the PLACES project which has received funding from the European Union's Horizon 2020 research and innovation programme under the Marie Skodowska-Curie grant agreement No 101086206, and by the MAIA project which has received funding from the European Union's Horizon 2020 research and innovation programme under the Marie Skodowska-Curie grant agreement No 951910. We acknowledge support from the Open Access Publishing Fund of the University of Tübingen. Carl Zeiss Vision International GmbH provided support in the form of salaries for authors PSD, SG, and SW. Eberhard Karls University Tuebingen provided support in the form of salary for author AA. University of Bologna provided support in the form of salaries for authors AB and PF. The funders had no role in study design, data collection and analysis, decision to publish, or preparation of the manuscript.

**Competing interests:** The authors have declared that no competing interests exist.

prehension movement strategies, highlighting the crucial relationship between visual feedback quality and object properties in the motor online control of the transport, manipulation and spatial components. Our results offer new insights into the implications of visual impairments on manual prehension movements.

## Introduction

Vision is a crucial process that enables us to perceive, interpret, and navigate our surroundings. It promotes the perception of various visual attributes such as shape, size, color, texture, and motion of objects, and therefore, it is fundamental in the development of fine motor skills and eye-hand coordination [1]. A comprehensive analysis of the vision capabilities and their interplay within the environment requires the evaluation of two key concepts: visual function, which encompasses the physiological mechanisms underlying visual perception, and functional vision, which pertains to the practical application of visual information in daily activities [2]. Visual function evaluation, both objectively and subjectively, typically involves exploring various aspects including visual acuity, contrast sensitivity, color vision, depth perception, and motion perception [3,4]. These aspects involve various facets which collectively determine our visual performance and, consequently, influence our functional daily capabilities [2,3,5,6]. In clinical practice, visual acuity (VA) and contrast sensitivity (CS) are perhaps the best-known and most frequently included measures, with VA being the most used [2]. While VA assesses the ability to identify fine details at a given distance and is particularly relevant for pattern recognition tasks such as reading [7,8], CS evaluates the ability to distinguish objects from their background as contrast decreases and is important for tasks such as driving or detecting changes in lighting environments [9]. As widely established, several factors, including age [10], refractive errors [11], and ocular diseases such as cataract, diabetic retinopathy, glaucoma, and age-related macular degeneration [12–16], among many others, can lead to a reduction in both VA and CS, further affecting motor functionality and quality of life [17].

Numerous studies have evaluated the impact of visual impairments on motor function [18–25]. In the context of reach-to-grasp activities, scientific evidence has shown that deficits in both central and peripheral visual fields can lead to a decrease in motor performance, specifically altering the transport and manipulation components of the reach-to-grasp movements [26–30]. In the early stages of these vision disorders, visual abilities may remain relatively unaffected, however, as the condition develops visual function declines resulting in a deterioration of functional vision, motor function, and therefore, quality of life [19]. In this context, VA and CS are two critical measures that provide valuable information over the course of such ocular disorders [31]. Indeed, the current scientific literature reports the impact of VA and CS losses on the reach-to-grasp movement accuracy [32–36]. However, there is a lack of studies examining how different levels of visual function loss affect motor output components during manual prehension tasks. This gap in vision research highlights the need for further exploration. Therefore, this study aims to provide new insights by

investigating the impact of different levels of visual function degradation, specifically in terms of VA and CS, on the reach and grasp components of manual prehension movements in young individuals with normal vision. We hypothesized that visual function degradation would impact the kinematic and spatial performance of prehension movements, with greater effects at higher degradation levels. We further expected these effects to be most evident during the deceleration phase of the movement, highlighting the importance of visual feedback quality in the online motor control of manual prehension actions. These findings could be particularly relevant for individuals with vision disorders where visual function is compromised. Understanding these effects will help us better comprehend how visual impairments could impact the motor output during manual prehension.

## Materials and methods

### Participants

A total of sixteen young adult subjects (7 women and 9 men) with a mean age of 31.11±3.03 years took part in the experiment. All participants were right-handed and free from any ocular disease. Those with recent eye surgeries, visual impairments, or conditions affecting the motor function of the arms and hands were excluded. Before starting the experiment, refractive errors were corrected for each participant by using trial lenses. The recruitment period was from 06 October 2021–30 June 2022.

The study was approved by the Ethics Committee at the Medical Faculty of the Eberhard Karls University Tuebingen and the University Hospital in Tuebingen. Prior to participation, all individuals provided written informed consent after being fully informed about the specifics, potential risks, and advantages of the study.

### Preliminary optometric examination

The refractive status of each participant was determined using a two-step process. In the first step, an objective evaluation was conducted using a wavefront aberrometer (i.Profiler plus, Carl Zeiss Vision GmbH, Aalen, Germany). In the second step, a subjective assessment was performed with a digital phoropter (Visuphor 500, Carl Zeiss Vision GmbH, Aalen, Germany). The subjective refractive correction was identified as the maximum plus lens power that gave the highest VA. To ensure the participants' refractive conditions were up to date for the study, the subjective correction was compared with their daily prescription. No modifications in their prescriptions were needed. Subsequently, during the experimental task, participants' habitual refraction was corrected using a trial frame and trial lenses (Oculus GmbH, Wetzlar, Germany).

### Visual conditions: Bangerter foils

Visual function degradation was induced using Bangerter occlusion foils (Breitfeld & Schliekert GmbH, Karben, Germany). These occlusion foils are flexible patches made of translucent plastic that attenuate both low and high spatial frequencies. Upon direct magnification, they exhibit a structure of not uniformly distributed scattering elements, as shown in Fig 1A.

For the current study, different degrees of visual degradation were implemented using five distinct densities: 0.8, 0.6, 0.3, 0.1, and <0.1. According to the manufacturer's specifications, these numerical designations indicate the degree to which the foil reduces VA. For instance, a density of 0.6 implies that the VA through such a foil would be approximately equivalent to 0.6 (decimal) or 20/30 (Snellen fraction). The lower the foil density number, the stronger the occlusion and therefore the higher the visual degradation (see Table 1 for further information on the VA values associated with each foil density). The occlusion foils were adhered to a Plano lens, which was subsequently placed on the right eye of the trial frame.

To establish a baseline for comparison, a Plano lens (no dioptric power) without any attached occlusion foil was used as a control condition. This approach facilitated the assessment of visual performance under normal viewing conditions, serving as a reference to evaluate the impact of the occlusion foils.

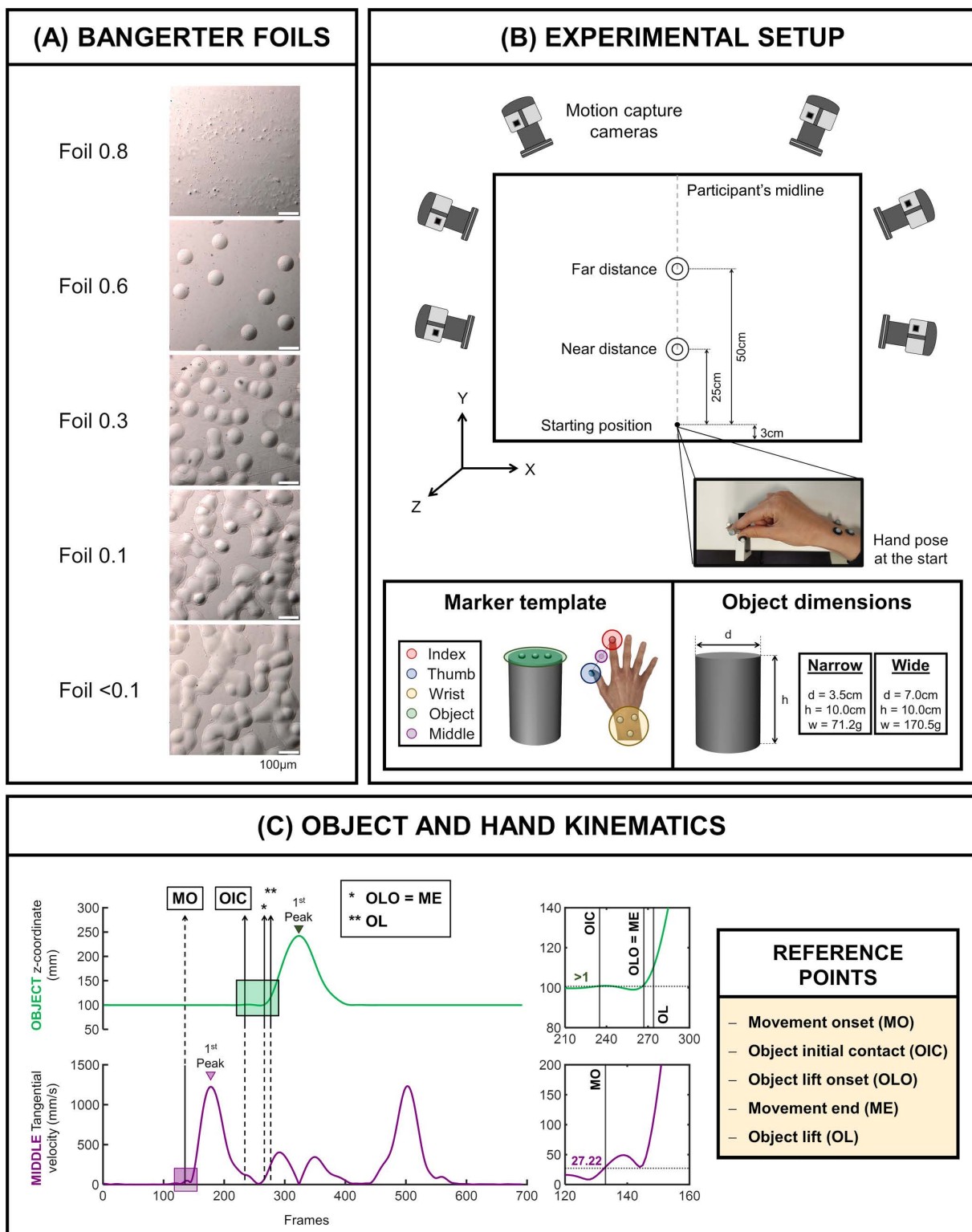

**Fig 1. Overview of visual conditions, experimental setup, and kinematic data analysis.** (A) Microscopic photographs of Bangerter occlusion foils captured at a 10x magnification. (B) Experimental Setup. The diagram illustrates the experimental setup with two object positions: a near distance of

25 cm and a far distance of 50 cm from the starting position, marked by a screw located 3 cm from the table edge. Hand and object trajectories were tracked by six infrared motion cameras. For that, five reflective markers were adhered to the dorsal surface of the hand and three markers on the object. Additionally, the three-dimensional average point between the index and thumb markers was calculated and referred to as middle (MID) marker. Two cylindrical objects were used as a target: narrow (diameter (d) = 3.5 cm; height (h) = 10.0 cm; weight (w) = 71.2 g) and wide (diameter (d) = 7.0 cm; height (h) = 10.0 cm; weight (w) = 170.5 g). (C) Object and hand kinematics. The upper plot, depicted in green, illustrates the z-coordinates of the object's central marker. The green box, expanded on the right, indicates where object initial contact, object lift onset, movement end, and object lift have been detected based on the object z-coordinate. The lower plot, represented in purple, shows the tangential velocity of the MID marker. The purple box, expanded on the right, represents where the movement onset has been located based on the MID marker tangential velocity. These values were used to determine five reference points essential for the computation of kinematic variables: movement onset (MO), object initial contact (OIC), object lift onset (OLO), movement end (ME) and object lift (OL). All data presented corresponds to a trial of a single participant.

**Table 1. Visual acuity values for each foil density according to the manufacturer's specifications in LogMAR, Decimal, Snellen (6 meters), and Snellen (20 feet) notations.**

| Bangerter occlusion foils | LogMAR | Decimal | Snellen (meter) | Snellen (feet) |
|---|---|---|---|---|
| Foil 0.8 | 0.10 | 0.80 | 6/7.5 | 20/25 |
| Foil 0.6 | 0.20 | 0.63 | 6/9.5 | 20/30 |
| Foil 0.3 | 0.50 | 0.32 | 6/19 | 20/60 |
| Foil 0.1 | 1.00 | 0.10 | 6/60 | 20/200 |
| Foil <0.1 | <1.00 | <0.10 | <6/60 | <20/200 |

**Objective characterization.** The imaging properties of a plano lens with attached Bangerter occlusion foils were characterized by acquiring their point spread function (PSF). To obtain each PSFs, the foils were illuminated with a monochromatic (wavelength, 532nm) and collimated beam. This beam, after passing through the foils, was focused onto a camera system (DMM37UX226, The Imaging Source GmbH, Bremen, Germany) using a converging lens with a focal length of 100 mm and a circular aperture of 5 mm in diameter. The PSFs were acquired through three specific zones within the circular aperture: at the geometric center, and at points 1.25 mm to the left and right from the geometric center.

The modulation transfer function (MTF) was computed from the PSF by applying the fast Fourier transform. Since PSFs were taken in three different zones per occlusion foil, the MTF values were averaged. The MTF accounts for the degradation of the image contrast at different spatial frequencies. The image quality through the Bangerter occlusion foils was objectively quantified by calculating the area under the MTF (AUMTF) up to 60 cycles per degrees (cpd). This was done using the following formula:

$$AUMTF = \sum_{i=1}^{M} \sum_{j=1}^{M} circ(i,j) \, MTF_{i,j} \, \Delta f^2$$

(1)

where: $MTF_{i,j}$ is the element $(i,j)$ of the $M \times M$ matrix that contains the MTF values; $circ(i,j)$ is a function that equals 1 for $\Delta f \sqrt{i^2 + j^2}$ <60 cpd and 0 otherwise; and $\Delta f$ is the pixel size of the MTF matrix in spatial frequency units. In this study, $\Delta f$ was set to 0.15 cpd.

Images of an optotype chart were captured through the occlusion foils to illustrate their impact on image properties (Fig 2C). The experimental setup included an optotype chart positioned at 64 cm from a lens with a focal length of 30 mm, a circular aperture with a diameter of 5 mm attached to the lens, and a camera system recording the chart image. The chart was illuminated by a white, fluorescent lamp. The camera's gamma value was set to 0.45 to mimic the non-linear intensity response of the human eye [37].

**Subjective characterization.** Bangerter occlusion foils were subjectively characterized by assessing the VA and CS of all participants across all foil density levels included in the experiment. VA was evaluated using a standardized adult Sloan Letter near vision chart (Good-Lite, Illinois, USA) in ETDRS format, which features five letters per row with equal spacing

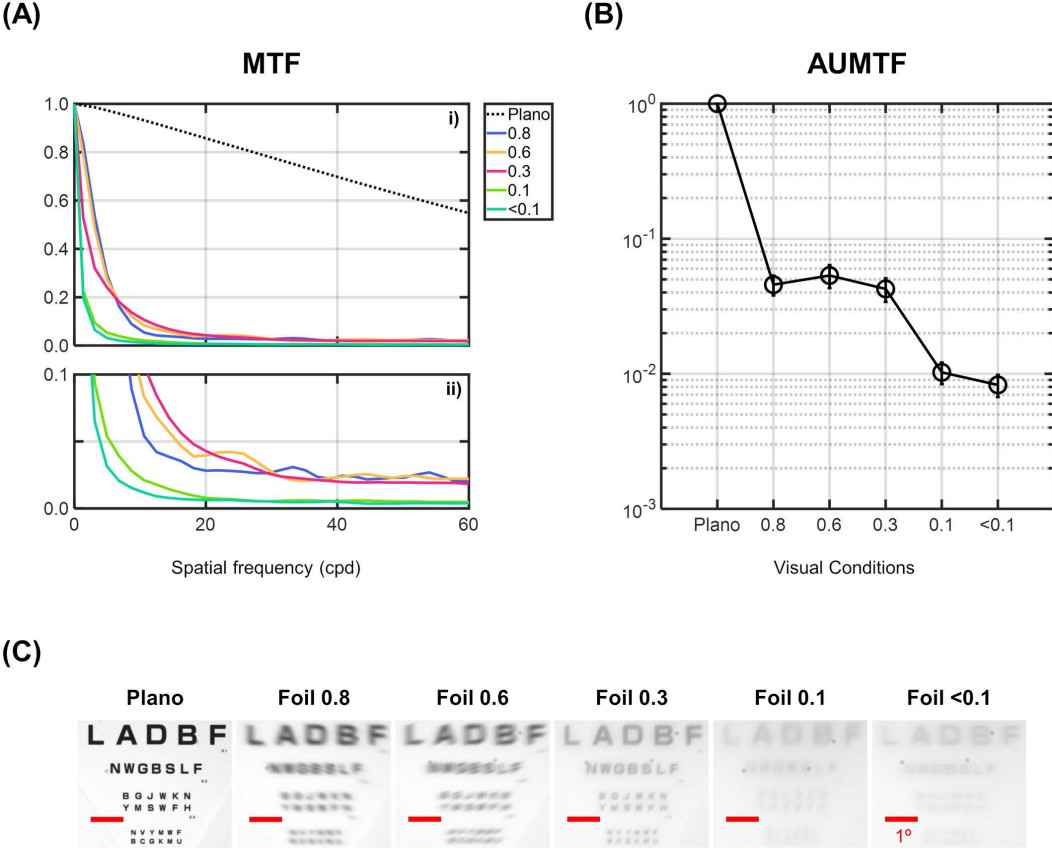

**Fig 2. Objective characterization of the different visual conditions.** (A) Radially averaged modulation transfer function (MTF), being plotted with a maximum range of: i) 1 and ii) 0.1. (B) Area under the MTF (AUMTF) for each visual condition. AUMTF values are normalized to the Plano condition. Error bars indicate standard deviations. (C) Images of an optotype chart acquired through the different visual conditions.

of rows and letters on a logarithmic scale. The line sizes varied between 0.05 and 2.0 in decimal scale (equivalent to 1.30 and −0.30 logMAR, respectively).

CS was measured using the Freiburg Vision Test (FrACT; version 3.10.5) [38], run by a laptop computer (Dell Latitude 5430, Dell Technologies Inc, Texas, USA) and displayed on a 9.7" monitor with a resolution of 2048x1536 pixels (Adafruit Qualia, Adafruit Industries, New York, USA). CS was evaluated using fixed-size Landolt C stimuli, which were composed of a combination of low and medium spatial frequencies and presented in four possible orientations (0°, 180°, 90°, 270°). Each participant completed a total of 24 trials under each visual condition. CS threshold was calculated using the Best PEST adaptive staircase procedure [39].

Both, VA, and CS tests were conducted monocularly (right eye) at a fixed distance of 40 cm. To maintain a constant viewing distance, both a forehead and chin rest were used.

## Experimental procedure

Participants were positioned in front of a white table, where they were asked to execute a reach-to-grasp task involving a real object under six different visual conditions. The object was cylindrical in shape, exhibiting a uniform gray hue and 10 cm high. Two specific diameters, 3.50 cm (narrow) and 7.00 cm (wide), along with their corresponding weights, 71.20

and 170.50 g, were employed. Furthermore, the object was placed at two distances along the participant's midline: 25 cm, referred to as the near distance, and 50 cm, referred to as the far distance (see Fig 1B for a detailed overview).

Each trial began with the participant's eyes closed and their right hand positioned at the starting position. After receiving the verbal cue "go", participants were instructed to open their eyes, visually locate the object, and reach and grasp the object using their index finger and thumb of their right hand. Once grasped, they were then required to lift it vertically and subsequently place it back on the table. The trial concluded with the participant bringing their hand back to the initial position. The starting position was marked by a metal screw located 3 cm from the table's edge in a central position with respect to the participant (Fig 1B). Participants were instructed to initiate and conclude each trial by grasping the screw with their index finger and thumb. They were encouraged to execute the reach-and-grasp action naturally, without specific instructions on finger placement for grasping the object or on where to subsequently position the object on the table. Between trials, participants were instructed to keep their eyes closed. To further ensure visual isolation, an opaque trial lens was positioned in front of the right eye to prevent participants from visually discerning the arrangement of the upcoming experimental condition. To verify task comprehension and reduce motor learning effects, a series of preliminary training trials were carried out before starting the main experiment [40].

In total, each participant conducted 72 trials, comprising a combination of 6 visual conditions, 2 object sizes, and 2 object distances, all repeated three times. The sequence of these trials was randomized to prevent task habituation and learning.

The experiment was conducted under monocular viewing conditions. Participants were required to wear a trial frame equipped with an opaque trial lens covering the left eye, while the occlusion foil was placed in front of the right eye. If necessary, trial lenses were incorporated to correct participants' refraction. Room illuminance was maintained at photopic levels (676.67 ± 13.72 lx). Right-hand dominance was confirmed using the Flinders Handedness survey [41] before the start of the experiment.

## Apparatus

Marker trajectory data were compiled at 100 Hz using a motion capture system consisting of a set of infrared cameras (Vicon Vero v2.2, Vicon Motion Systems Ltd., Oxford, UK) which registered the three-dimensional position and orientation of the participants' hand during the experimental task. Five retro-reflective markers (6 mm diameter) were attached on the dorsal surface of the participants' right hand: one on the thumb nail, one on the index fingernail and three on the wrist (Fig 1B). Three additional markers were placed on the top of each object to track its movement (Fig 1B). The three-dimensional average point between the index and thumb markers was calculated and referred to as the middle (MID) marker. The MID marker was used as a reference for calculating spatial kinematic variables and to minimize potential biases from experimental conditions. To ensure accurate marker detection, each experimental session started with the optical camera calibration using the Active Wand, a T-shaped electronic motion capture calibration device (Active Wand, Vicon Motion Systems Ltd., Oxford, UK). The calibration tool was actively moved within the desired capture area until the end of the calibration process, which was defined by the registration of 1000 frames of valid Wand data. Subsequently, the global coordinate system was established, determining the origin of the capture volume, and thus the center and orientation (x, y, z). Nexus software (Nexus 2.10, Vicon Motion Systems Ltd., Oxford, UK) was employed to control the different system components, encompassing camera calibration processes, data acquisition, storage, visualization, and the exportation of three-dimensional data related to each trial.

## Data processing and kinematic parameters

For the three-dimensional data processing, three phases were established: (1) reconstruction and labeling, (2) gap-filling, and (3) data filtering. Firstly, all grayscale blobs and unlabeled markers generated by unwanted reflections were removed from all recorded trials to eliminate potential artifacts introduced during the recording phase. Additionally, each marker was labeled with its corresponding identifier, facilitating accurate tracking throughout the data processing pipeline. Secondly,

we filled the gaps in frames with trajectory breaks by using the positions and orientations of the markers detected within the three-dimensional space throughout the trials. And thirdly, the three-dimensional data were filtered using a fourth-order Butterworth low-pass filter with a cutoff frequency of 6 Hz.

The impact of visual degradation on prehension movements was assessed through the analysis of different kinematic parameters. For that, five temporal reference points were established: movement onset (MO), movement end (ME), object initial contact (OIC), object lift onset (OLO), and object lift (OL) (Fig 1C). MO was defined as the first frame in which the middle (MID) marker tangential velocity exceeded 27.22 mm/s and consistently remained above it until the first peak of the MID marker velocity. This threshold was determined based on the percentage difference in tangential velocity between the MID and index markers, aligning it with an index finger velocity of 30 mm/s. OIC was characterized as the first time point at which the object z-axis went > 1 mm. ME and OLO were both determined as the first frame in which object position was moved vertically (z-axis) more than 1 mm and remained above this threshold until the first z-axis peak position. This ensured the conclusion of the reaching phase and a successful object grasping. This alignment in time of both reference points has also been used by other authors [42–44]. OL was denoted as the first frame when the object's position was shifted vertically more than 10 mm and continued to rise until it reached its maximum peak.

A total of 25 parameters were quantified (see Table 2 for abbreviations and definitions). As seen in Table 2, these parameters were grouped into four main categories: General Kinematics, Reach Dynamics, Grasp Dynamics, and Spatial Kinematics. General Kinematics assessed the overall movement efficiency. Reach Dynamics parameters offered a comprehensive understanding of the transport components of the reaching phase. Grasp Dynamics revealed the adjustments made by the index finger and thumb to grasp the object, examining the precision and coordination of the manipulation components. Spatial Kinematics evaluated the spatial components of the movement, analyzing how the hand navigated through space during the experimental task. Collectively, all these parameters facilitated a comprehensive and detailed analysis of the impact of visual function degradation on the reach and grasp components of manual prehension movements. Note that these parameters have been previously documented and validated for the analysis of reach-to-grasp movements [30,32–34,45,46].

Among all the mentioned parameters in Table 2, it is important to highlight that grip aperture was defined as the Euclidean distance between the index finger and thumb markers. As in our last study [47], to calculate the real grip aperture, which represents the distance between the inner surfaces of the index finger and thumb, a 5-second measurement was performed by each subject during which they were instructed to bring their fingerprints into contact with each other. The distance between the index finger and thumb markers recorded was then named as "snap grip aperture". The real grip aperture was therefore calculated frame-by-frame as the difference between the distance of the index-thumb markers and the mean value of the "snap grip aperture".

The time course of the tangential velocity and grip aperture was also analyzed based on the experimental conditions. Each variable was normalized to the duration of the movement and divided into 1%-time bins, with 0% representing the movement onset and 100% representing the movement end.

The impact of the experimental conditions on reaching trajectories was also assessed. To achieve this, deviations in the x, y, and z components of the MID marker were calculated relative to a virtual straight trajectory. This virtual straight trajectory was defined as the straight line connecting the x, y, and z coordinates of the MID marker at the movement onset frame to the x, y, and z coordinates at the movement end frame. Deviations for each component were determined by comparing the actual trajectory to the virtual straight trajectory. To ensure consistency and minimize variability, deviations were normalized to movement duration and divided into 1%-time bins, with 0% representing the movement onset and 100% representing the movement end. The influence of the experimental conditions was then examined based on the temporal progression of deviations in each of the x, y, and z components.

Custom MATLAB algorithms (The MathWorks, Massachusetts, United Stated) were developed for data filtering, processing, and analysis.

**Table 2. Definitions and abbreviations of the kinematic parameters.**

| Kinematic parameter (abbreviation) | Definition |
|---|---|
| **Reference points** | |
| Movement Onset (MO) | First-time frame at which the tangential velocity of the middle marker became greater than 27.22 mm/s and remained above this value until the first peak velocity. |
| Object Initial Contact (OIC) | First-time frame at which the object marker was displaced by more than 1 mm in the z-coordinate. |
| Object Lift Onset (OLO) | First-time frame at which the object marker was displaced by more than 1 mm in the z-coordinate and continued to rise over time. |
| Movement End (ME) | First-time frame at which the object marker was displaced by more than 1 mm in the z-coordinate and continued to rise over time (equivalent to Object Lift Onset). |
| Object Lift (OL) | First-time frame at which the object marker was displaced by more than 10 mm in the z-coordinate and continued to rise over time. |
| **General Kinematics** | |
| Movement Duration (MD) | Time from movement onset to movement end. |
| **Reach Dynamics** | |
| Average Velocity (AV) | Mean velocity from movement onset to movement end. |
| Peak Velocity (PV) | Maximum middle marker velocity. |
| Peak Acceleration (PA) | Maximum middle marker acceleration. |
| Peak Deceleration (PD) | Maximum middle marker deceleration. |
| % Peak Velocity (% PV) | Percentage of the movement at which the peak velocity is reached. |
| % Peak Acceleration (% PA) | Percentage of the movement at which the peak acceleration is reached. |
| % Peak Deceleration (% PD) | Percentage of the movement at which the peak deceleration is reached. |
| Time to Peak Velocity (ttPV) | Time from movement onset to maximum middle marker velocity. |
| Time to Peak Acceleration (ttPA) | Time from movement onset to peak acceleration. |
| Time to Peak Deceleration (ttPD) | Time from movement onset to peak deceleration. |
| Time from PD to OIC (tPD–OIC) | Time from peak deceleration to object initial contact. |
| Deceleration time (DecT) | Calculated by subtracting the time to peak velocity from movement duration. |
| Normalized deceleration time (% DecT) | Calculated by subtracting the time to peak velocity from movement duration and dividing this difference by the movement duration. |
| Time spent at low Velocity (tLV) | Calculated by subtracting the time to peak deceleration from movement duration. |
| Normalized time spent at low velocity (% tLV) | Calculated by subtracting the time to peak deceleration from movement duration and dividing this difference by the movement duration. |
| **Grasp Dynamics** | |
| Average Grip Aperture (AGA) | Mean grip aperture between thumb and middle finger from movement onset to movement end. |
| Peak Grip Aperture (PGA) | Maximum grip aperture between thumb and middle finger before object contact. |
| % Peak Grip Aperture (% PGA) | Percentage of the movement at which the peak grip aperture is reached. |
| Time to peak Grip Aperture (ttPGA) | Time from movement onset to maximum grip aperture. |
| Time from peak Grip Aperture to object lift onset (tPGA-OLO) | Time from maximum grip aperture to object lift onset. |
| Time Post contact (tPC) | Time from object initial contact to object lift. |
| **Spatial kinematics** | |
| Path length (PL) | Length of the middle trajectory from movement onset to movement end. |
| Maximum lateral deviation (MLD) | Maximum lateral deviation (x-axis) of the middle trajectory, from movement onset to movement end, compared to a straight trajectory. |
| Maximum vertical deviation (MVD) | Maximum height (z-axis) of the middle trajectory from movement onset to movement end. |

## Statistical analysis

The statistical analysis was conducted using the MATLAB R2023a statistical toolbox (The MathWorks, Massachusetts, United States) and SPSS statistical software package, version 29.0 for Windows (SPSS, Chicago, Illinois, USA). The normality assumption was assessed using the Shapiro-Wilk test. Since the data followed a normal distribution, appropriate statistical tests were applied, including three-way repeated measures ANOVA and paired t-tests. A three-way repeated measures ANOVA was used to examine the interaction between the three independent variables (foil density, object size, and object distance) on the various kinematic variables. Mauchly's test of sphericity was used to assess whether the assumption of sphericity was met. In cases where the sphericity assumption was violated, Greenhouse-Geisser or Huynh-Feldt corrections were applied based on the epsilon value. To account for multiple comparisons, the Bonferroni correction was applied. Paired t-tests were employed to compare the temporal evolution of variables such as velocity, grip aperture, and trajectory deviations under the different experimental conditions. To correct for multiple comparisons across time, the Benjamini-Hochberg procedure was applied to control the false discovery rate (FDR) at an alpha < 0.05. A significance level of $p < 0.05$ was used for all statistical tests.

## Results

### Objective characterization of Bangerter foils

Fig 2A shows the radially averaged MTF for Plano and all Bangerter occlusion foils as a function of spatial frequency. Plano condition achieved the highest MTF values across the different spatial frequencies when compared to the occlusion foils. It is evident that the occlusion foils caused a rapid decline in the MTF values as the spatial frequency increased. In general, the MTF values decreased as the foil density number decreased, with the < 0.1 foil showing the lowest values.

Fig 2B displays the averaged AUMTF values for all visual conditions. Consistent with the MTF observations, AUMTF values decreased as the foil density number decreased. Considering Plano condition as a reference, a significant reduction was noted when comparing its AUMTF values with those of the occlusion foils. Following this, the reduction in AUMTF values was more gradual, with a moderate decrease among the 0.8, 0.6, and 0.3 foil densities. For the 0.1 and <0.1 foils, there was a more pronounced drop in AUMTF values, with <0.1 foil exhibiting the lowest values and poorest image quality.

The impact of these metrics can be visualized in the images acquired under the different visual conditions (Fig 2C). Subjectively, the 0.8 and 0.6 foils appear to provide similar blur levels. The image produced by the 0.3 foil looks sharper but contains lower spatial frequencies compared to those generated by the 0.8 and 0.6 foils. As expected, the contrast levels are perceived to be lower with the 0.1 and <0.1 foils, but the details of the optotype chart are still visible. The presence of such details in the image is due to the MTF values provided by the 0.1 and <0.1 foils being very low but not zero (as shown in Fig 2A).

### Subjective characterization of Bangerter foils

On average, all Bangerter foil densities significantly reduced VA compared to the Plano condition ($p < 0.001$ for all). Specifically, the mean VA was −0.10 ± 0.00 logMAR for Plano, 0.20 ± 0.03 logMAR for 0.8 foil, 0.08 ± 0.05 logMAR for 0.6 foil, 0.20 ± 0.07 logMAR for 0.3 foil, 0.47 ± 0.06 logMAR for 0.1 foil, and 0.64 ± 0.10 logMAR for the < 0.1 foil (Fig 3). Compared to Plano condition, VA was reduced by 49.20% with the 0.8 foil, 33.25% with the 0.6 foil, 48.65% with the 0.3 foil, 72.85% with the 0.1 foil, and 81.35% with the < 0.1 foil. Comparing foils stepwise, mean VA decreased as the foil density increased, with the lowest VA values observed for the highest density (<0.1) foil (Fig 3; note that values are in LogMAR). However, this trend was not consistent for the 0.8 and 0.6 foils since the mean VA was significantly worse when looking through the 0.8 foil compared to 0.6 ($p < 0.001$; Fig 3).

Similar findings were observed for CS, which also decreased as the foil density increased. As seen in Fig 3, the mean CS values were found to be the highest for the Plano condition, 1.76 ± 0.08 Log units. For the Bangerter foils, the mean

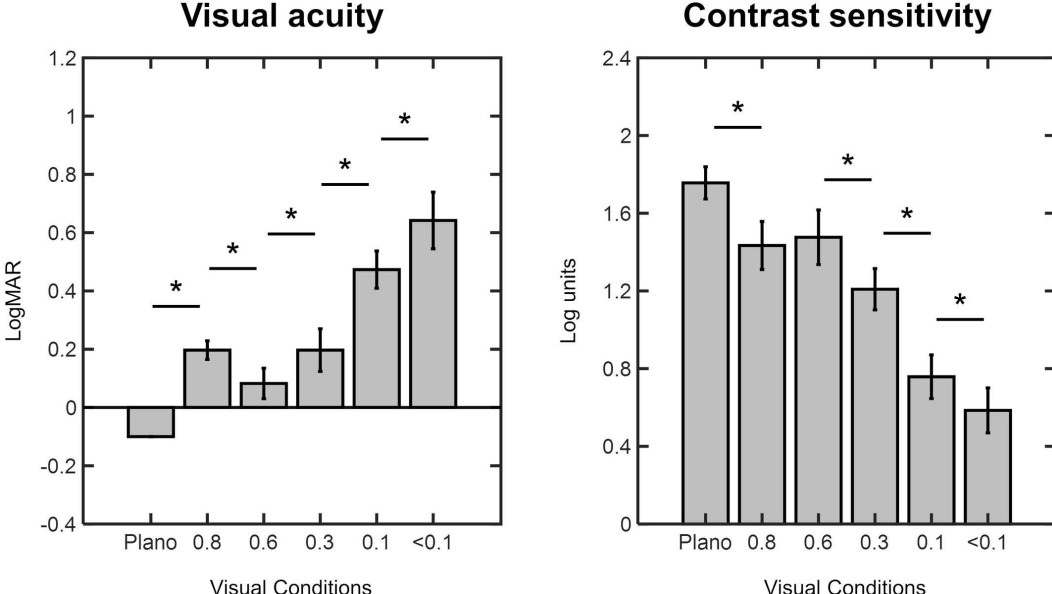

**Fig 3. Visual acuity (in LogMAR) and contrast sensitivity (in Log units) across different experimental visual conditions.** Error bars represent standard deviations. Asterisks indicate p values < 0.05.

values were as follows: 1.43 ± 0.12 Log units for 0.8 foil, 1.48 ± 0.14 Log units for 0.6 foil, 1.21 ± 0.11 Log units for 0.3 foil, 0.76 ± 0.11 Log units for 0.1 foil, and 0.59 ± 0.12 Log units for the < 0.1 foil (Fig 3). CS was significantly reduced under all foil densities compared to the Plano condition (p < 0.001 for all). This reduction represented a decrease in CS of 18.36% with the 0.8 foil, 15.94% with 0.6 foil, 31.17% with 0.3 foil, 56.83% with 0.1 foil, and 66.69% with < 0.1 foil. Nevertheless, the CS reduction was not consistent across all foil densities, since the 0.8 and 0.6 foils produced similar values (p = 0.39; Fig 3).

### Kinematics

S1–S8 Figs show the mean values for all kinematic parameters across all visual conditions: Plano, 0.8, 0.6, 0.3, 0.1, and < 0.1, based on the object's size and distance. Additionally, S1 Table presents the main statistical effects of the experimental conditions and their interactions on the kinematic parameters.

**General kinematics. Movement duration (MD):** MD was significantly influenced by the foil density (F(5, 235) = 10.92, p < 0.001, $\eta_p^2$ = 0.19; S1 Table). The greater the reduction in visual function, the longer the MD (S1 Fig). Specifically, the longest MDs were found under foil densities of 0.1 and < 0.1. The MD associated with the 0.1 foil density was significantly greater than those observed under the Plano, 0.8, 0.6, and 0.3 conditions (p ≤ 0.02; S1 Fig). Similarly, MD under the < 0.1 foil was significantly higher than those under the Plano and 0.8 conditions (p ≤ 0.01; S1 Fig). Plano condition, when compared to 0.8 and 0.6 foil densities, did not result in significantly different MDs (p ≥ 0.48; S1 Fig).

MD also showed a significant increase with increasing object size (F(1, 47) = 5.95, p = 0.02, $\eta_p^2$ = 0.11; S1 Table). On average, the wide object generated significantly longer durations. There was also a significant interaction effect between foil density and object size on MD (F(4.56, 214.43) = 4.27, p < 0.001, $\eta_p^2$ = 0.08; S1 Table). When comparing MDs between both object sizes across different foil densities, the wide object resulted in significantly longer durations than the narrow object under the Plano, 0.8, and 0.3 conditions (p ≤ 0.03; S1 Fig). However, no significant differences between sizes were found under the 0.6, 0.1, and < 0.1 foil densities (p ≥ 0.08; S1 Fig). When assessing MD across different foil densities for

each object size separately, significant differences were observed only for the narrow object. In particular, the 0.1 and <0.1 foils resulted in significantly longer durations compared to the other visual conditions (p ≤ 0.01; S1 Fig). In contrast, the wide object did not generate any significant variation in the MD across the different foil densities (p ≥ 0.05; S1 Fig).

MD also revealed a significant increase with increasing object distance (F(1, 47) = 947.16, p < 0.001, $\eta_p^2$ = 0.95; S1 Table). As expected, longer distances resulted in longer durations (S1 Fig). However, there was no significant interaction effect between foil density and object distance on MD (F(5, 235) = 1.68, p < 0.14, $\eta_p^2$ = 0.03; S1 Table).

**Reach dynamics. Average velocity (AV):** AV significantly decreased as foil density increased (F(4.54, 213.24) = 8.43, p < 0.001, $\eta_p^2$ = 0.15; S1 Table). The highest AVs were observed under the Plano condition (S2 Fig). The 0.3, 0.1, and <0.1 foils exhibited significantly lower AVs compared to Plano (p < 0.01 for all; S2 Fig). Additionally, AV was significantly affected by object distance (F(1, 47) = 1486.94, p < 0.001, $\eta_p^2$ = 0.97; S1 Table). Greater distances resulted in higher velocities (S2 Fig and S1 Table).

**Peak velocity (PV) and peak acceleration (PA):** PV and PA were significantly impacted by foil density ((F(5, 235) = 3.49, p < 0.01, $\eta_p^2$ = 0.07) and (F(5, 235) = 3.12, p = 0.01, $\eta_p^2$ = 0.06) respectively; S1 Table). As visual degradation increased, both PV and PA decreased (S2 Fig). Compared to the Plano condition, PV values were significantly reduced under the 0.6, 0.1 and <0.1 foils (p = 0.04, p = 0.02, and p = 0.04, respectively). However, for PA values, significant reductions were observed under the <0.1 foil (p = 0.04), but not under the 0.6 and 0.1 foils (p = 0.057, and p = 0.054, respectively).

Object distance also influenced both PV (F(1, 47) = 860.08, p < 0.001, $\eta_p^2$ = 0.95; S1 Table) and PA (F(1,47) = 45.19, p < 0.001, $\eta_p^2$ = 0.49; S1 Table). On average, participants achieved higher PVs and PAs when reaching the object located at far (p < 0.001 for both; S2 Fig).

The percentage of the movement at which these peaks were reached was also affected. In general, PV was reached at approximately one-third of the total movement (32.08 ± 0.50%, S3 Fig), which was influenced by the interaction effect between foil density and object size (F(5, 235) = 46.65, p = 0.01, $\eta_p^2$ = 0.06; S1 Table). As seen in S3 Fig, compared to the narrow object, PVs were reached earlier with the wide object under the Plano and 0.8 conditions (p = 0.02 both), but not under 0.6, 0.3, 0.1 and <0.1 foils (p ≥ 0.06). Additionally, when evaluating the percentage of the movement at which the PVs were reached across different foil densities for each object size separately, significant differences were found only for the narrow object (S3 Fig). PV was reached significantly earlier under the 0.1 foil compared to Plano and 0.8 conditions (p < 0.01 and p = 0.02, respectively; S3 Fig). In contrast, the wide object did not generate any significant variation in the PVs across the different foil densities (p ≥ 1.00; S3 Fig).

Furthermore, PA was achieved at 24.95 ± 1.03% of the total movement. Here, the timing of PA was only significantly influenced by the object distance (F(1, 47) = 70.31, p < 0.001, $\eta_p^2$ = 0.60; S1 Table). As illustrated in S3 Fig, on average, PAs were reached earlier at the far distance (20.67 ± 0.54%) compared to the near distance (29.23 ± 1.23%).

**Deceleration phase:** The time participants spent in the deceleration phase was influenced by the foil density (F(4.30, 202.02) = 7.42, p < 0.001, $\eta_p^2$ = 0.14; S1 Table). As seen in S5 Fig, stronger foil densities resulted in longer deceleration time (DecT). In general, participants spent significantly more time decelerating with 0.1 and <0.1 foils compared to Plano and 0.8 conditions (p < 0.03 for all).

Object size also affected DecT (F(1, 47) = 5.04, p = 0.03, $\eta_p^2$ = 0.10; S1 Table), with longer DecTs observed for the wide object. Additionally, there was a significant interaction effect between foil density and object size on the time participants took to decelerate (F(5, 235) = 5.07, p < 0.001, $\eta_p^2$ = 0.10; S1 Table). The comparison of DecT values between both object sizes revealed that the wide object generated significantly longer DecTs under Plano, 0.8, and 0.3 conditions (p ≤ 0.02; S5 Fig). In contrast, 0.6, 0.1 and <0.1 foils led to similar DecTs for both object sizes (S5 Fig). When comparing DecTs across different foil densities for each object size separately, significant differences were found only for the narrow object, and not for the wide object (S5 Fig). The 0.1 and <0.1 foils resulted in significantly longer DecTs compared to Plano, 0.8, 0.6, and 0.3 visual conditions (p ≤ 0.04; S5 Fig). The wide object showed no significant variations in DecT values across the different foil densities (p = 1.0; S5 Fig).

 

DecT also increased significantly with object distance ($F(1, 47) = 403.61$, $p < 0.001$, $\eta_p^2 = 0.90$; S1 Table). Far distance led to longer DecTs across all foil densities ($F(5, 235) = 2.43$, $p = 0.04$, $\eta_p^2 = 0.05$; S1 Table).

Peak deceleration (PD) values were similarly affected by object distance ($F(1, 47) = 34.37$, $p < 0.001$, $\eta_p^2 = 0.42$; S1 Table), with higher PD values observed when participants reached objects located at far distance (S2 Fig).

Moreover, the normalized deceleration time (% DecT) was influenced by the interaction effect between foil density and object size ($F(5, 235) = 46.65$, $p = 0.01$, $\eta_p^2 = 0.06$; S1 Table). As observed in S3 Fig, % DecT was statistically higher when participants reached for the wide object under the Plano and 0.8 conditions ($p = 0.02$ both), but not under the 0.6, 0.3, 0.1, and <0.1 foils ($p > 0.06$).

In the final portion of the deceleration phase, from the PD to the end of the movement, similar findings were observed. Foil density ($F(4.44, 208.52) = 7.96$, $p < 0.001$, $\eta_p^2 = 0.15$; S1 Table), and object distance ($F(1, 47) = 364.59$, $p < 0.001$, $\eta_p^2 = 0.89$; S1 Table) significantly affected the time spent at low velocity (tLV). Stronger foil densities, and far distance led participants to perform the task at low velocity for a longer period (S5 Fig). In addition, a significant interaction effect between foil density and object size on tLV was found ($F(5, 235) = 4.12$, $p = 0.001$, $\eta_p^2 = 0.08$; S1 Table). When comparing tLV values between both object sizes, the wide object produced significantly longer tLV under the Plano, 0.8, and 0.3 conditions ($p \leq 0.04$; S5 Fig). However, no significant differences between sizes were observed under the 0.6, 0.1, and <0.1 foil densities ($p \geq 0.07$; S5 Fig). When assessing tLV across different foil densities for each object size separately, significant differences were observed only for the narrow object (S5 Fig). The 0.1 and <0.1 foils resulted in significantly longer tLV compared to Plano, 0.8, 0.6, and 0.3 visual conditions ($p \leq 0.02$; S5 Fig). The wide object showed no significant variations in tLV values across the different foil densities ($p \geq 0.37$; S5 Fig).

**Grasp dynamics. Average grip aperture (AGA) and peak grip aperture (PGA):** AGA values were larger when participants executed the task under stronger foil densities ($F(5, 235) = 43.54$, $p < 0.01$, $\eta_p^2 = 0.07$; S1 Table), with the wide object ($F(1, 47) = 2386.54$, $p < 0.001$, $\eta_p^2 = 0.98$; S1 Table) and at far distance ($F(1, 47) = 64.58$, $p < 0.001$, $\eta_p^2 = 0.58$; S1 Table). Additionally, on AGA, there were significant interaction effects among foil density and object size ($F(5, 235) = 6.38$, $p < 0.001$, $\eta_p^2 = 0.12$; S1 Table), foil density and object distance ($F(5, 235) = 3.57$, $p < 0.01$, $\eta_p^2 = 0.07$; S1 Table) and foil density, object size and object distance ($F(4.48, 210.42) = 4.61$, $p < 0.001$, $\eta_p^2 = 0.09$; S1 Table). These results align with the findings observed for the PGA, which was also influence by foil density ($F(4.44, 208.52) = 9.50$, $p < 0.001$, $\eta_p^2 = 0.17$; S1 Table), object size ($F(1, 47) = 742.55$, $p < 0.001$, $\eta_p^2 = 0.94$; S1 Table), the interaction among foil density and object size ($F(5, 235) = 3.66$, $p < 0.01$, $\eta_p^2 = 0.07$; S1 Table) and the interaction among foil density, object size and object distance ($F(5, 235) = 5.48$, $p < 0.001$, $\eta_p^2 = 0.10$; S1 Table). As shown in S6 Fig, the effect of foil density on both AGA and PGA was more evident when participants performed the task with the narrow object at a far distance.

The timing of the appearance of the PGA revealed interesting findings. On average, the PGA was reached at $61.81 \pm 0.70\%$ of the movement (S6 Fig). The percentage of the movement at which the PGA was reached (% PGA) was influenced by object distance ($F(1, 47) = 20.09$, $p < 0.001$, $\eta_p^2 = 0.30$; S1 Table) and the interaction effect between foil density and object size ($F(5, 235) = 4.34$, $p < 0.001$, $\eta_p^2 = 0.09$; S1 Table). Comparing both sizes, PGA was reached earlier with the narrow object under the 0.1 and <0.1 foils ($p < 0.04$ both; S6 Fig), but not under Plano, 0.8 0.6, and 0.3 conditions ($p \geq 0.25$; S6 Fig). Additionally, when participants reached for the narrow object, PGA occurred significantly earlier under the 0.1 foil compared to Plano, 0.8, 0.6, and 0.3 conditions ($p < 0.02$; S6 Fig).

The time from MO to PGA (ttPGA) was significantly longer with higher foil density ($F(4.33, 203.37) = 4.21$, $p < 0.01$, $\eta_p^2 = 0.08$; S1 Table), wide object ($F(1, 47) = 18.70$, $p < 0.001$, $\eta_p^2 = 0.29$; S1 Table), and far distance ($F(1, 47) = 463.66$, $p < 0.001$, $\eta_p^2 = 0.91$; S1 Table). The highest ttPGA values were found under 0.1 and <0.1 foil densities (S7 Fig). On average, participants required 30 ms less time to reach the PGA under the Plano condition compared to the 0.1 and <0.1 foil conditions ($p < 0.05$; S7 Fig).

Furthermore, the time from PGA to OLO (tPGA-OLO) revealed similar trends. tPGA-OLO was influenced by foil density ($F(5, 235) = 6.21$, $p < 0.001$, $\eta_p^2 = 0.12$; S1 Table), object distance ($F(1, 47) = 72.77$, $p < 0.001$, $\eta_p^2 = 0.61$; S1 Table), and the

interaction effect between foil density and object size ($F_{(5, 235)} = 5.71$, $p < 0.001$, $\eta_p^2 = 0.11$; S1 Table). A greater foil density produced higher tPGA-OLO values. The effect of foil density on tPGA-OLO was more noticeable when participants performed the experimental task with the narrow object at a far distance (S7 Fig).

**Spatial kinematics.** **Visual function degradation also impacted the spatial trajectory:** Participants took longer PL trajectories when performing the task with stronger foil densities ($F_{(5, 235)} = 4.51$, $p < 0.001$, $\eta_p^2 = 0.09$; S1 Table). Further analysis revealed that this was driven by wider lateral trajectories (MLD variable; $F_{(3.57, 167.97)} = 3.22$, $p < 0.02$, $\eta_p^2 = 0.06$; S1 Table and S8 Fig) and higher vertical trajectories (MVD variable; $F_{(5, 235)} = 6.94$, $p < 0.001$, $\eta_p^2 = 0.13$; S1 Table and S8 Fig).

PL was also influenced by the interaction among foil density, object size and object distance presenting the stronger effects with the narrow size at far distance ($F_{(5, 235)} = 2.73$, $p = 0.02$, $\eta_p^2 = 0.06$; S1 Table and S8 Fig). These outcomes suggest that the participants adjusted their reaching spatial trajectory based on the degradation of their visual function. The following sections further explore these findings.

**Time course of MID marker velocity and grip aperture.** **MID marker velocity:** Fig 4 shows the time course of the MID marker velocity under each of the visual conditions as a function of the different combinations of object sizes and distances. With respect to Plano condition, regardless of object size and distance, foil densities of 0.3, 0.1 and <0.1 exhibited significantly lower velocities within a span of 40% to 100% of the movement ($p < 0.05$; Fig 4). Prior to the PV, from onset up to ~30% of the movement, no significant differences were detected in any of the comparisons between the Plano condition and foil densities.

Considering object size and distance, similar trends were observed. For the narrow object at near distance, compared to Plano, 0.1, and <0.1 foils caused a significant reduction in the MID marker velocity, ranging from 40% to 80% of the movement ($p < 0.05$; Fig 4). For the narrow object at far distance, following the PV, compared to the Plano condition, 0.1, and <0.1 foils showed significantly lower velocities, ranging from 35% to 99% of the movement ($p < 0.05$; Fig 4). Interestingly, within the acceleration phase (2–23% of the movement), the 0.1 foil resulted in significantly higher velocity ($p < 0.05$; Fig 4). For the wide object at near distance, the MID marker velocity was significantly impacted before the PV. From 2% to 22% of the movement, the Plano condition showed higher velocities compared to those observed under the 0.1 foil ($p < 0.05$; Fig 4). At far distance, the 0.6 foil was the only one that induced significant velocity discrepancies in the MID marker velocity compared to the Plano condition. Prior to PV, the 0.6 foil led to significantly higher velocities between 12% and 20% of the movement ($p < 0.05$; Fig 4), whereas following PV, it resulted in significantly lower velocities from 33% to 92% of the movement ($p < 0.05$; Fig 4).

Fig 5 presents the MID marker velocity across different visual conditions. For each visual condition, comparisons were made between near and far distances for both narrow and wide objects, as well as between narrow and wide objects for both distances, near and far. When comparing velocities between both distances within each visual condition (Fig 5A), statistically higher velocities were found at far distance compared to near distance. This finding was consistent for both narrow and wide object sizes across all visual conditions up to 60% of the movement ($p < 0.05$; Fig 5A).

Examining the velocities within each visual condition as a function of object size (Fig 5B) revealed interesting results. At both near and far distances, the narrow object generated significantly greater velocities compared to the wide object under the Plano, 0.8, 0.6, and 0.3 conditions. These differences were found after 80% of the completed movement ($p < 0.05$; Fig 5B).

**Grip aperture:** Fig 6 shows the comparison of the time course of the grip aperture values under each of the visual conditions according to object sizes and distances.

Considering object size and distance, the narrow size at far position had the most significant impact on grip aperture values (Fig 6). Relative to the Plano condition, under the 0.1, and <0.1 foils, participants over-scaled their grip aperture within the range 40–60% of the movement ($p < 0.05$; Fig 6). Within this interval, the mean grip aperture values under the Plano condition was $62.15 \pm 1.79$ mm, whereas under the 0.1 and <0.1 foils, it increased to $68.73 \pm 1.65$ mm and $68.16 \pm 1.68$ mm, respectively (Fig 6).

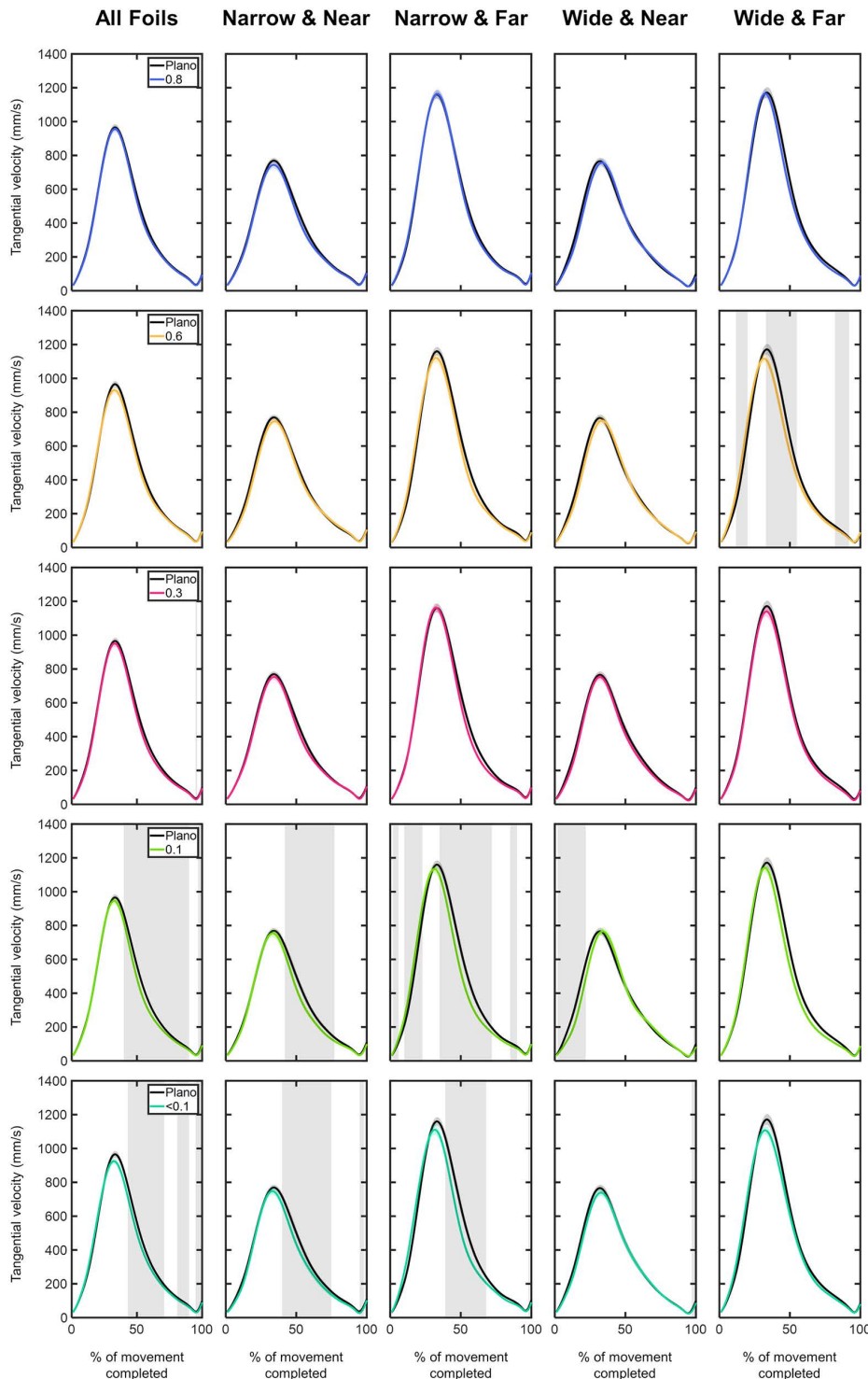

**Fig 4. Comparison of the middle (MID) marker tangential velocity profiles (in mm/s) between Plano and each Bangerter foil across the different experimental conditions (object size and object distance).** Tangential velocity values are plotted as a function of the percentage of movement completed, ranging from 0% (movement onset) to 100% (movement end). The values are represented by the mean of all trials from all participants, with shaded areas indicating the standard error of the mean. Grey shaded areas indicate statistically significant differences (FDR-corrected $p$ values < 0.05) in the comparison.

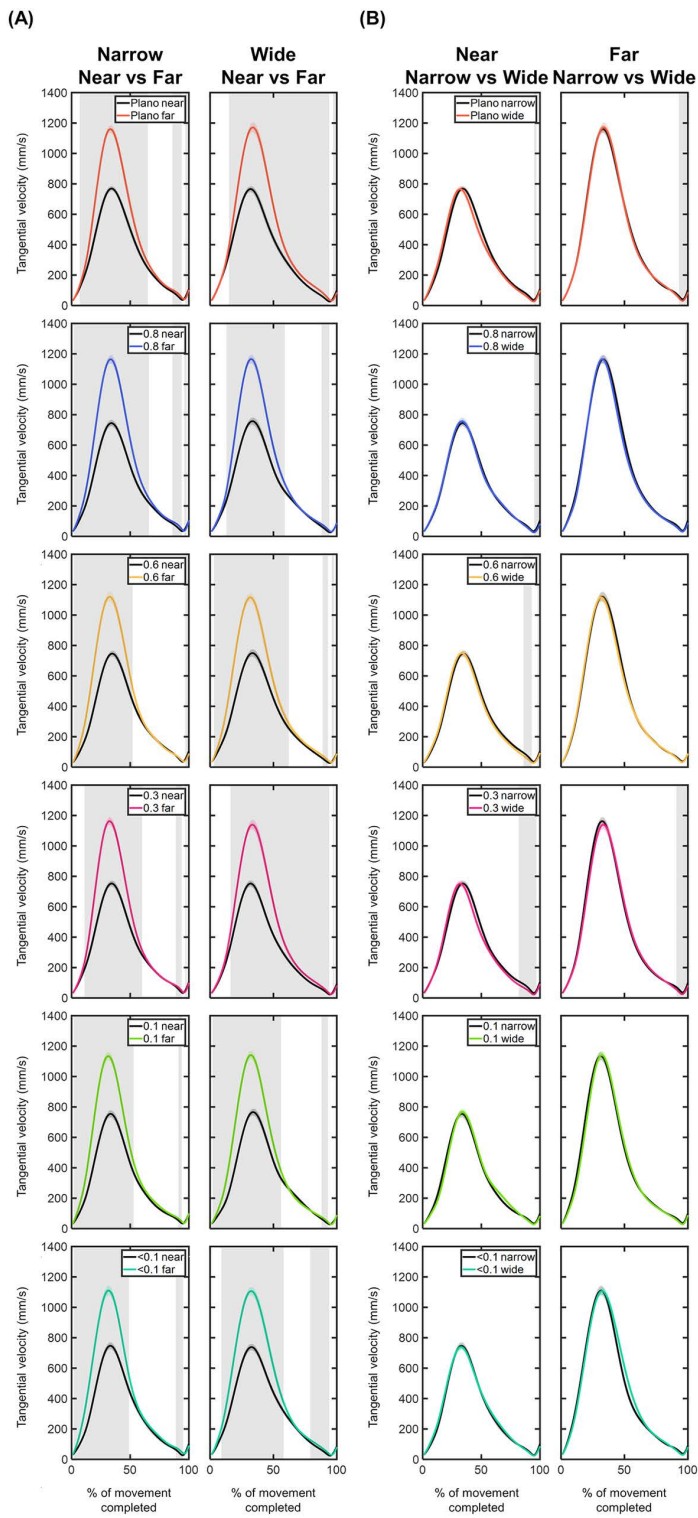

**Fig 5. Comparison of the middle (MID) marker tangential velocity profiles (in mm/s) within each visual condition based on (A) object distance and (B) object size.** Tangential velocity values are plotted as a function of the percentage of movement completed, ranging from 0% (movement onset) to 100% (movement end). The values are represented by the mean of all trials from all participants, with shaded areas indicating the standard error of the mean. Grey shaded areas indicate statistically significant differences (FDR-corrected p values < 0.05) in the comparison.

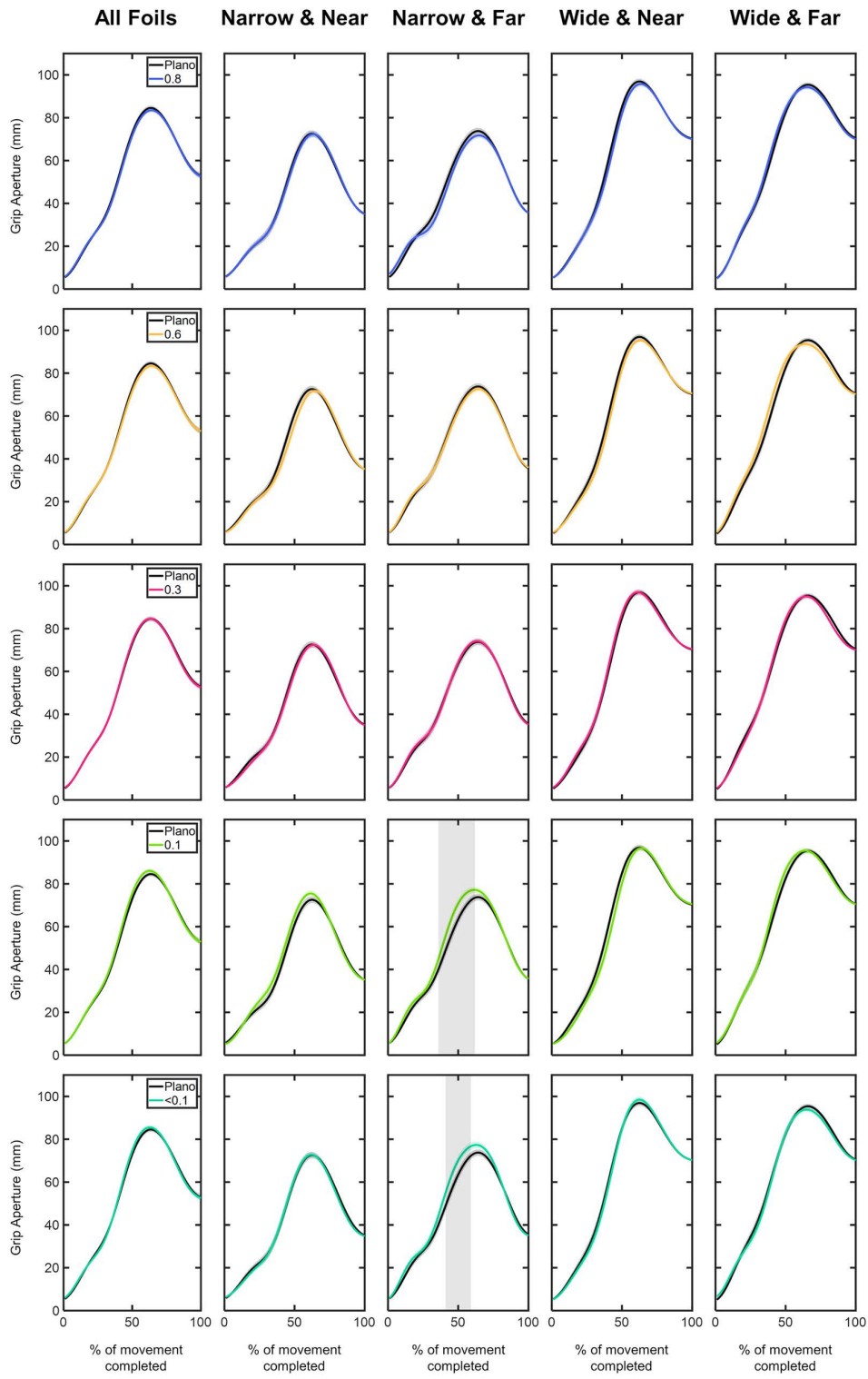

**Fig 6. Comparison of the grip aperture profiles (in mm) between Plano and each Bangerter foil across the different experimental conditions (object size and object distance).** Grip aperture values are plotted as a function of the percentage of movement completed, ranging from 0% (movement onset) to 100% (movement end). The values are represented by the mean of all trials from all participants, with shaded areas indicating the standard error of the mean. Grey shaded areas indicate statistically significant differences (FDR-corrected p values < 0.05) in the comparison.

Fig 7 presents the time course of grip aperture values for each visual condition as a function of object size and distance. For each visual condition and both narrow and wide sizes, the far distance showed significantly higher grip aperture values than the near distance during movement intervals both before and after the PGA (p < 0.05; Fig 7A). For the narrow size, these trends were consistent across all visual conditions, except for the 0.6 and <0.1 foils (Fig 7A). The 0.6 foil triggered significantly higher grip aperture values within the first half of the movement. However, the <0.1 foil led to larger grip aperture values throughout almost the entire movement course (6–94%; p < 0.05; Fig 7A). For the wide size, the far distance showed significantly higher values in the intervals both before and after the PGA across all visual conditions except for the 0.3 foil (5–50% and 70–95%; p < 0.05; Fig 7A).

As expected, within each visual condition, the comparison between narrow and wide sizes revealed that, across all visual conditions and object distances, grip aperture values were significantly larger for the wide object (p < 0.05; Fig 7B). However, this trend did not remain consistent throughout the entire course of the movement. As shown in Fig 7B, statistically significance discrepancies in grip aperture values between narrow and wide sizes started to emerge after 12% of the movement was completed. The onset of these statistical differences showed variations depending on both the visual conditions and object distance. For instance, strongest density foils exhibited differences later in time. Specifically, at near distance, significant differences were observed starting at 22% of the completed movement for the Plano, 0.8, and 0.6 conditions, while for the 0.1 and <0.1 foils, these differences emerged from 29% of the movement. Similarly, at far distance, significant differences started from 17% of the movement under Plano, 0.8 and 0.6 conditions, while for the 0.3, 0.1, and <0.1 foils, they appeared from 24% (p < 0.05; Fig 7B).

**Trajectory deviations of the MID marker – x, y, z components.  X component:** The time course of the MID marker trajectory deviations in the x component can be seen in Figs 8 and 9. Overall, the x trajectory showed a rightward deviation with respect to the straight trajectory, reaching its peak around 50% of the movement. After this peak, the x trajectory gradually realigned with the straight trajectory until the end of the movement (Fig 8). Compared to the Plano condition, no significant x deviations were observed for any foil (p > 0.05; Fig 8). The largest x deviations were found under the 0.1 and <0.1 foils for the narrow object positioned at far distance, with maximum differences of 4.57 mm and 5.22 mm occurring within the 20–50% of the movement.

For each visual condition, the comparison of x deviations based on object distance and size revealed significant findings (Fig 9). For the narrow size, the far-positioned object resulted in significantly higher x deviations under the 0.1 and <0.1 foils for nearly the entire range of movement (9–99% of the movement; p < 0.05; Fig 9A). The comparison between both sizes showed larger x deviations for the wide object under 0.1 and <0.1 foils at near distance (40–80% of the movement; p < 0.05; Fig 9B).

**Y component:** Figs 10 and 11 show the time course of the trajectory deviations of the MID marker in the y component. Similar deviation profiles were observed across all conditions. In the initial phase of the movement, up to 25%, the trajectory showed a negative deviation, indicating that the y trajectory of the MID marker trailed behind the corresponding point of the straight trajectory. From 25% of the completed movement, the y trajectory exhibited positive deviation values, meaning the y trajectory exceeded its corresponding point on the straight trajectory. The peak of y deviation was observed around 55% of the completed movement (Fig 10). On average, the most positive y deviation values were: 115.64 ± 3.19 mm for Plano; 118.66 ± 3.37 mm for 0.8 foil; 118.79 ± 3.49 mm for 0.6 foil; 118.92 ± 3.29 mm for 0.3 foil; 120.34 ± 3.34 mm for 0.1 foil; and 120.16 ± 3.42 mm for <0.1 foil (Fig 10).

Compared to the Plano condition, y deviations induced by visual degradation were most pronounced for the narrow object positioned at far distance (10–99% of the movement; p < 0.05; Fig 10).

Comparing deviations between both distances and both sizes revealed consistent trends across all visual conditions (Fig 11). The far-positioned object showed significantly more negative y deviations within the first 25% of the movement and greater positive y deviations after 30% of the movement (p < 0.05; Fig 11A). Across all visual conditions, no significant differences in y-axis deviation were observed between both object sizes at any distance (p > 0.05; Fig 11B).

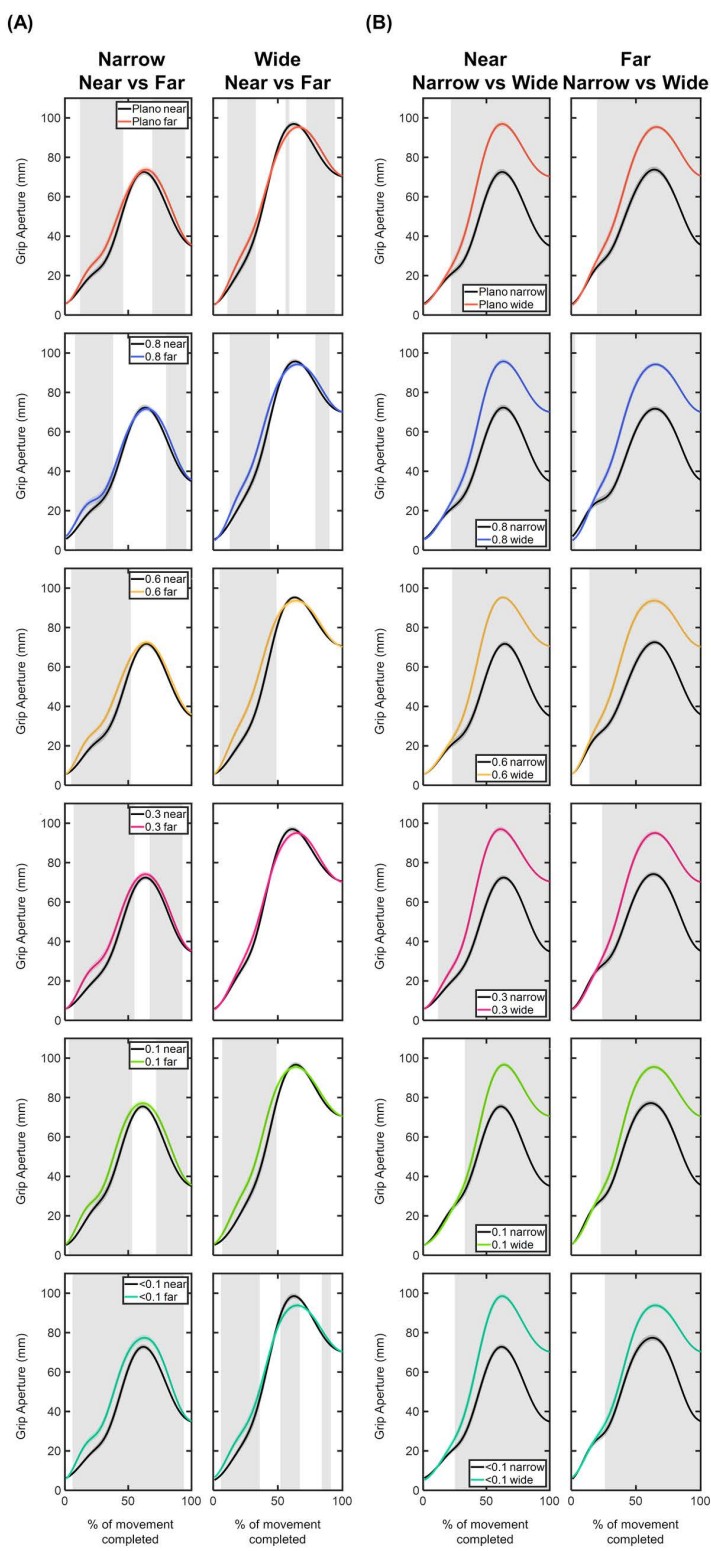

**Fig 7. Comparison of the grip aperture profiles (in mm) within each visual condition based on (A) object distance and (B) object size.** Grip aperture values are plotted as a function of the percentage of movement completed, ranging from 0% (movement onset) to 100% (movement end). The values are represented by the mean of all trials from all participants, with shaded areas indicating the standard error of the mean. Grey shaded areas indicate statistically significant differences (FDR-corrected p values < 0.05) in the comparison.

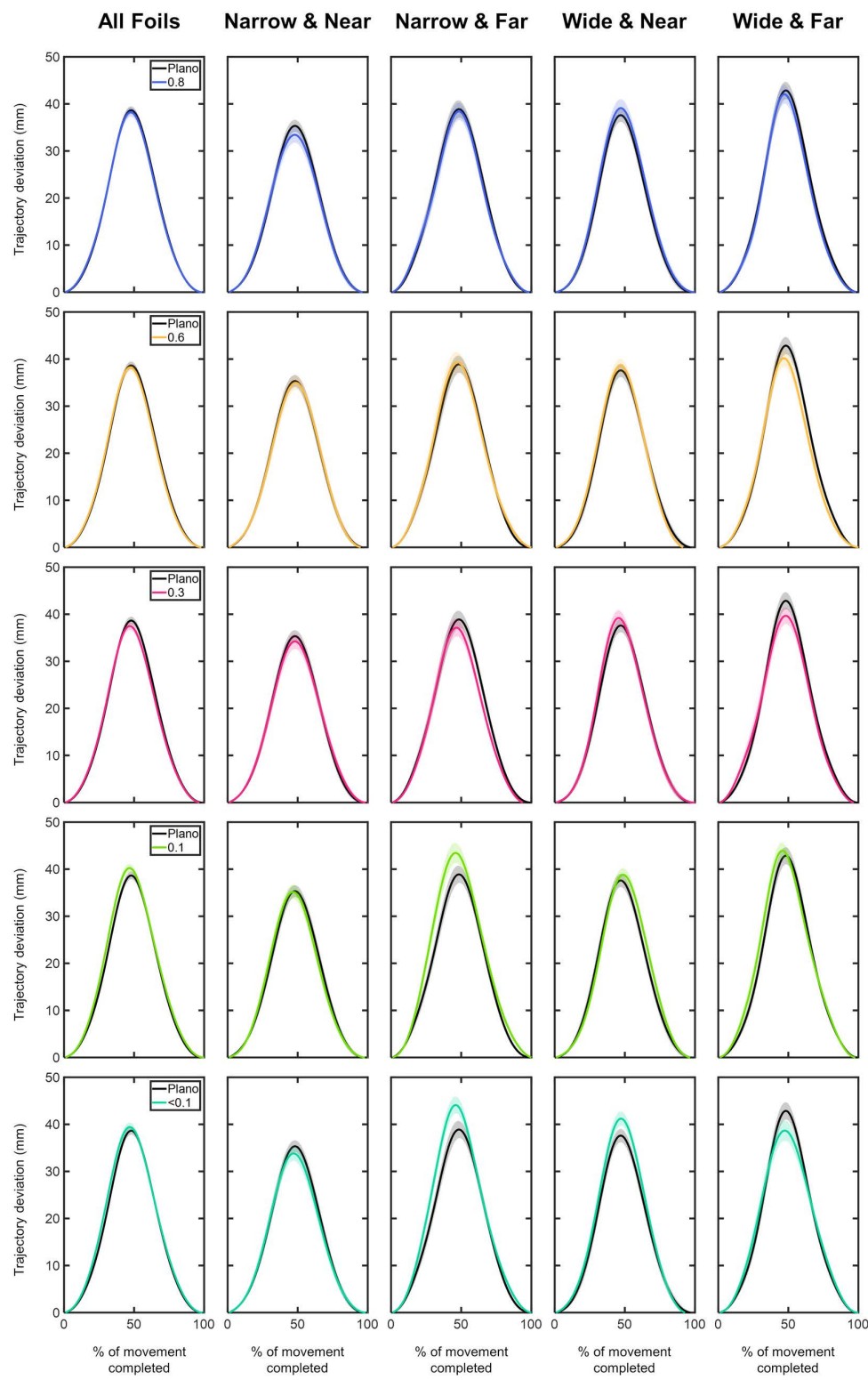

**Fig 8. Comparison of the x-component deviations (in mm) of the middle (MID) marker trajectory between the Plano condition and each Bangerter foil across different experimental conditions (object size and object distance).** Trajectory deviation values are plotted as a function of the percentage of movement completed, ranging from 0% (movement onset) to 100% (movement end). Negative values indicate leftward deviations, while positive values represent rightward deviations. The values are represented by the mean of all trials from all participants, with shaded areas indicating the standard error of the mean. Grey shaded areas indicate statistically significant differences (FDR-corrected p values < 0.05) in the comparison.

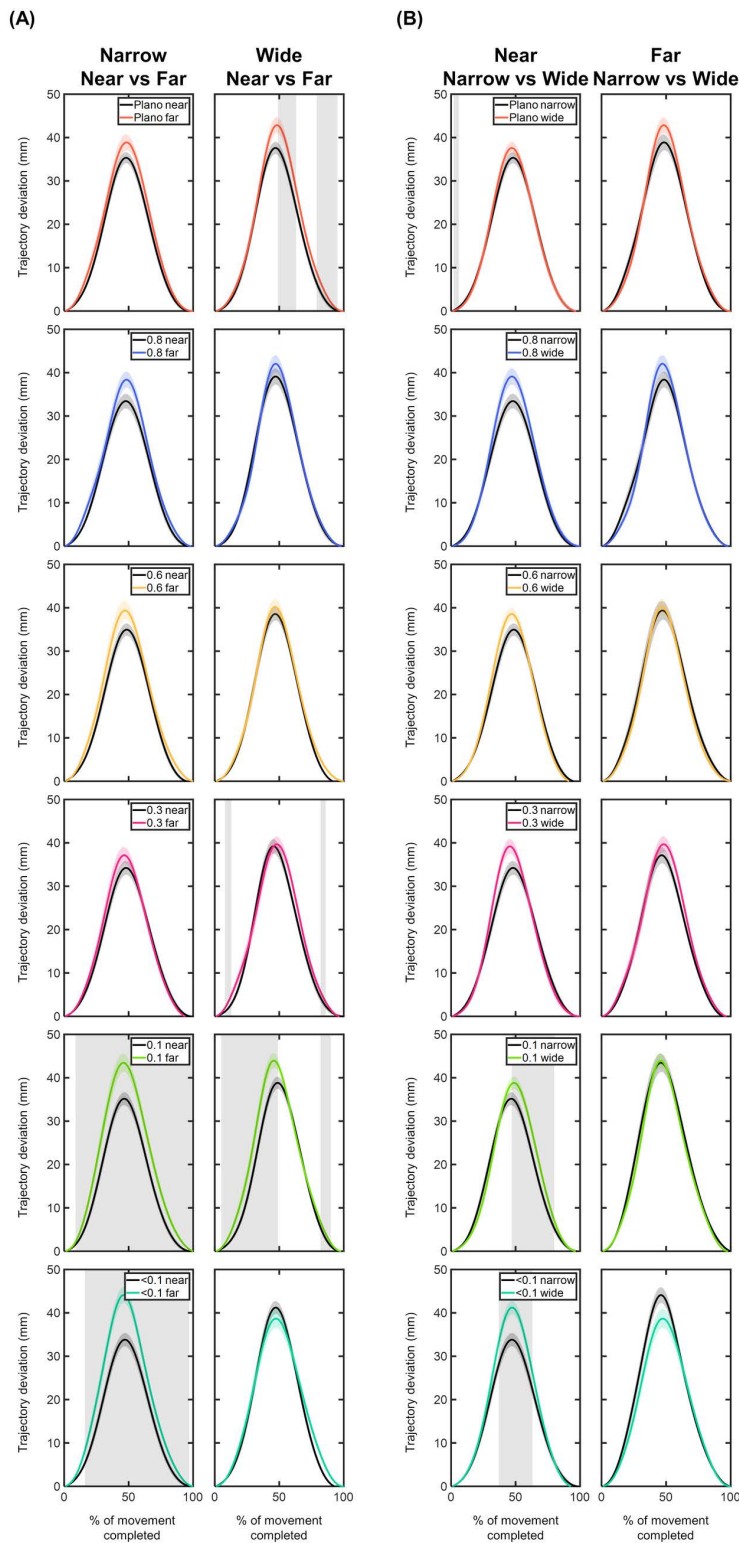

**Fig 9. Comparison of the x-component deviations (in mm) of the middle (MID) marker trajectory within each visual condition based on (A) object distance and (B) object size.** Trajectory deviation values are plotted as a function of the percentage of movement completed, ranging from 0% (movement onset) to 100% (movement end). Negative values indicate leftward deviations, while positive values represent rightward deviations. The values are represented by the mean of all trials from all participants, with shaded areas indicating the standard error of the mean. Grey shaded areas indicate statistically significant differences (FDR-corrected p values < 0.05) in the comparison.

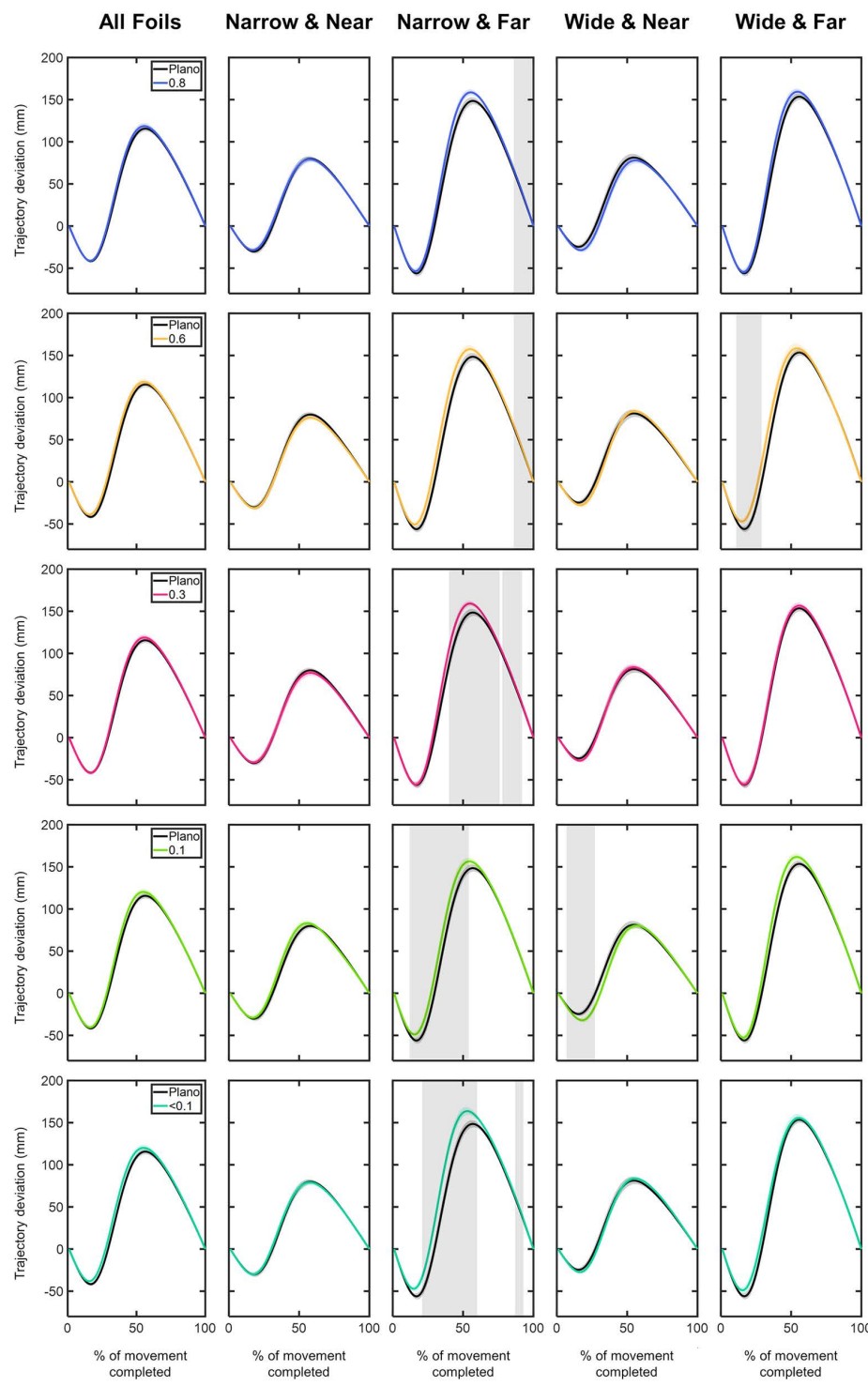

**Fig 10. Comparison of the y-component deviations (in mm) of the middle (MID) marker trajectory between the Plano condition and each Bangerter foil across different experimental conditions (object size and object distance).** Trajectory deviation values are plotted as a function of the percentage of movement completed, ranging from 0% (movement onset) to 100% (movement end). Negative values indicate the y-component trajectory was behind the straight trajectory, while positive values indicate it exceeded the straight trajectory. The values are represented by the mean of all trials from all participants, with shaded areas indicating the standard error of the mean. Grey shaded areas indicate statistically significant differences (FDR-corrected p values < 0.05) in the comparison.

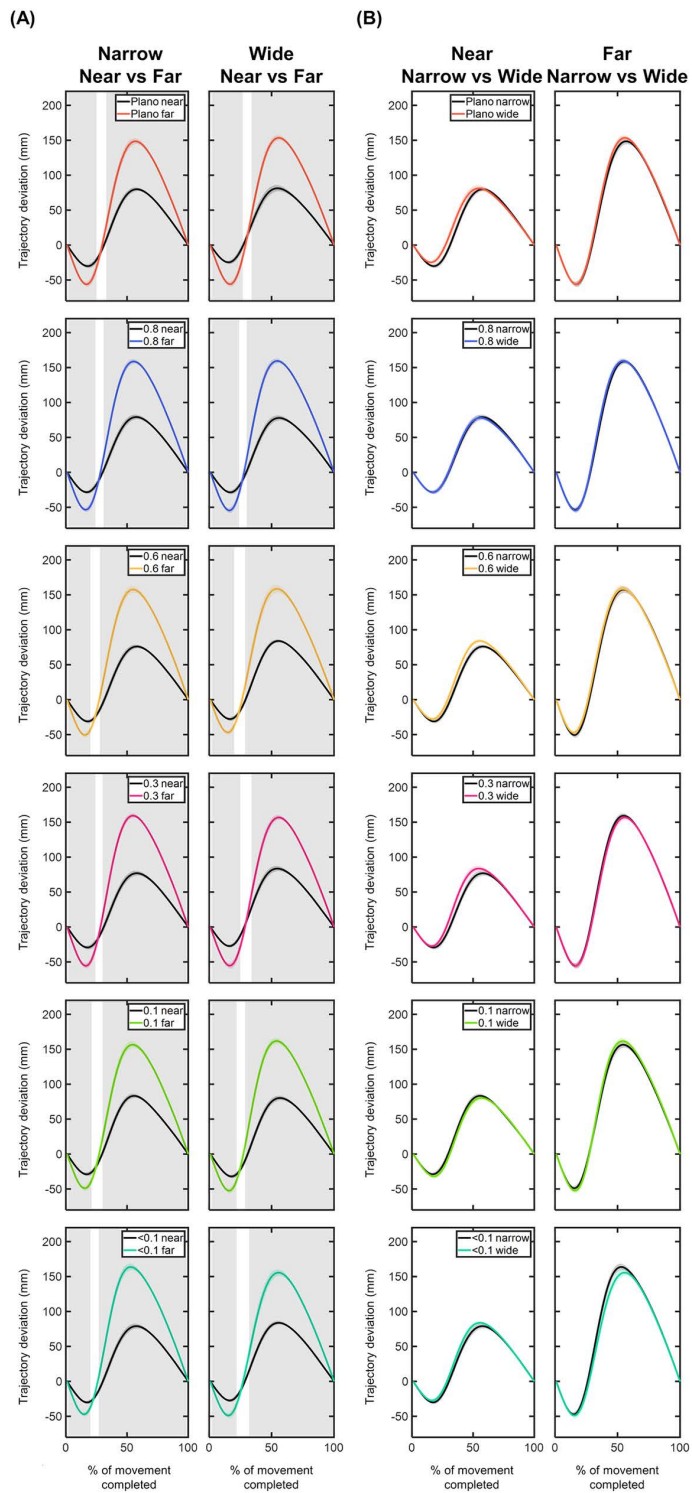

**Fig 11. Comparison of the y-component deviations (in mm) of the middle (MID) marker trajectory within each visual condition based on (A) object distance and (B) object size.** Trajectory deviation values are plotted as a function of the percentage of movement completed, ranging from 0% (movement onset) to 100% (movement end). Negative values indicate the y-component trajectory was behind the straight trajectory, while positive values indicate it exceeded the straight trajectory. The values are represented by the mean of all trials from all participants, with shaded areas indicating the standard error of the mean. Grey shaded areas indicate statistically significant differences (FDR-corrected p values < 0.05) in the comparison.

**Z component:** The z component trajectory deviations of the MID marker are presented in Figs 12 and 13. Overall, the z deviations demonstrated positive values, peaking at approximately 40% of the movement (Fig 12). On average, the maximum z deviation values were: $60.36 \pm 1.76$ mm for Plano; $56.57 \pm 1.61$ mm for 0.8 foil; $57.30 \pm 1.68$ mm for 0.6 foil; $58.10 \pm 1.77$ mm for 0.3 foil; $62.72 \pm 1.76$ mm for 0.1 foil; and $62.18 \pm 1.84$ mm for <0.1 foil (Fig 12). In particular, the highest z deviation values were observed for the 0.1 and <0.1 foils under the narrow size object located at far distance: $73.51 \pm 3.74$ mm and $74.35 \pm 3.75$ mm, respectively (Fig 12).

Compared to the Plano condition, the 0.8, 0.3, and <0.1 foils led to significantly lower z deviation values between 40–80% of the movement ($p < 0.05$; Fig 12). Interestingly, when participants reached for the narrow object at far distance, the 0.1 and <0.1 foils resulted in significantly greater z-axis deviations during the first half of the movement (10–40%; $p < 0.05$; Fig 12).

For each visual condition and both object sizes, the far distance resulted in significantly higher z deviations within the first (2–45%) and second half (from 63% onwards) of the movement course ($p < 0.05$; Fig 13A).

The comparison between sizes within each visual condition revealed differences. At near distance, the wide size showed significantly larger z deviations in the 0.8, 0.6, 0.3, and <0.1 foils (25–55% interval; $p > 0.05$; Fig 13B). In contrast, at far distance, the narrow size produced significantly higher z deviations in the 0.1, and <0.1 foils (4–88% interval; $p < 0.05$; Fig 13B).

## Discussion

The current study aimed to evaluate the impact of different levels of visual function degradation on the reach and grasp components of prehension movements in normal-sighted individuals. For this purpose, participants were instructed to perform natural reach-to-grasp movements on real objects of different sizes and spatial positions, while their visual function was altered by viewing through different densities of Bangerter occlusion foils. Our findings revealed four key aspects. First, the visual function degradation significantly impacted on the kinematic and spatial performance of prehension movements during the evaluated task. As hypothesized, the reported effects were dependent on the level of visual degradation applied. Reductions greater than 70% in VA and 55% in CS generated the greatest impact on the prehension components. Second, the time course analysis of the reach, grasp, and spatial components of prehension movements showed that subtle reductions greater than 30% in VA and 15% in CS were sufficient to lead participants to adopt compensatory strategies. Third, the effects of visual function degradation on the transport, manipulation, and spatial components of prehension movements were influenced by the intrinsic (size) and extrinsic (spatial position) object properties. Fourth, although the influence of object size and distance on the prehension components is well established, our observations suggest that the level of visual function degradation can modulate this influence.

The general findings of the current study will be summarized in four main sections (visual function, reach dynamics, grasp dynamics, and spatial kinematics) and its implications will be further addressed in a final remarks section.

### Visual function

The optical and visual function degradation resulting from each of the Bangerter occlusion foils was evaluated through both objective and subjective characterization.

The optical properties were objectively characterized and compared though the MTF and AUMTF of each foil. As expected, the non-degraded visual condition (Plano) exhibited the highest MTF and AUMTF values compared to the occlusion foils, demonstrating higher image contrast and quality. The <0.1 foil showed the lowest MTF and AUMTF values, indicating the poorest image quality.

The subjective degradation of visual function by the Bangerter occlusion foils was evaluated through measures of VA and CS. Both metrics revealed that visual function was reduced by all foil densities compared to non-degraded vision (Plano). The highest foil density resulted in reductions of up to 81% in VA and 67% in CS. Interestingly, as reported in

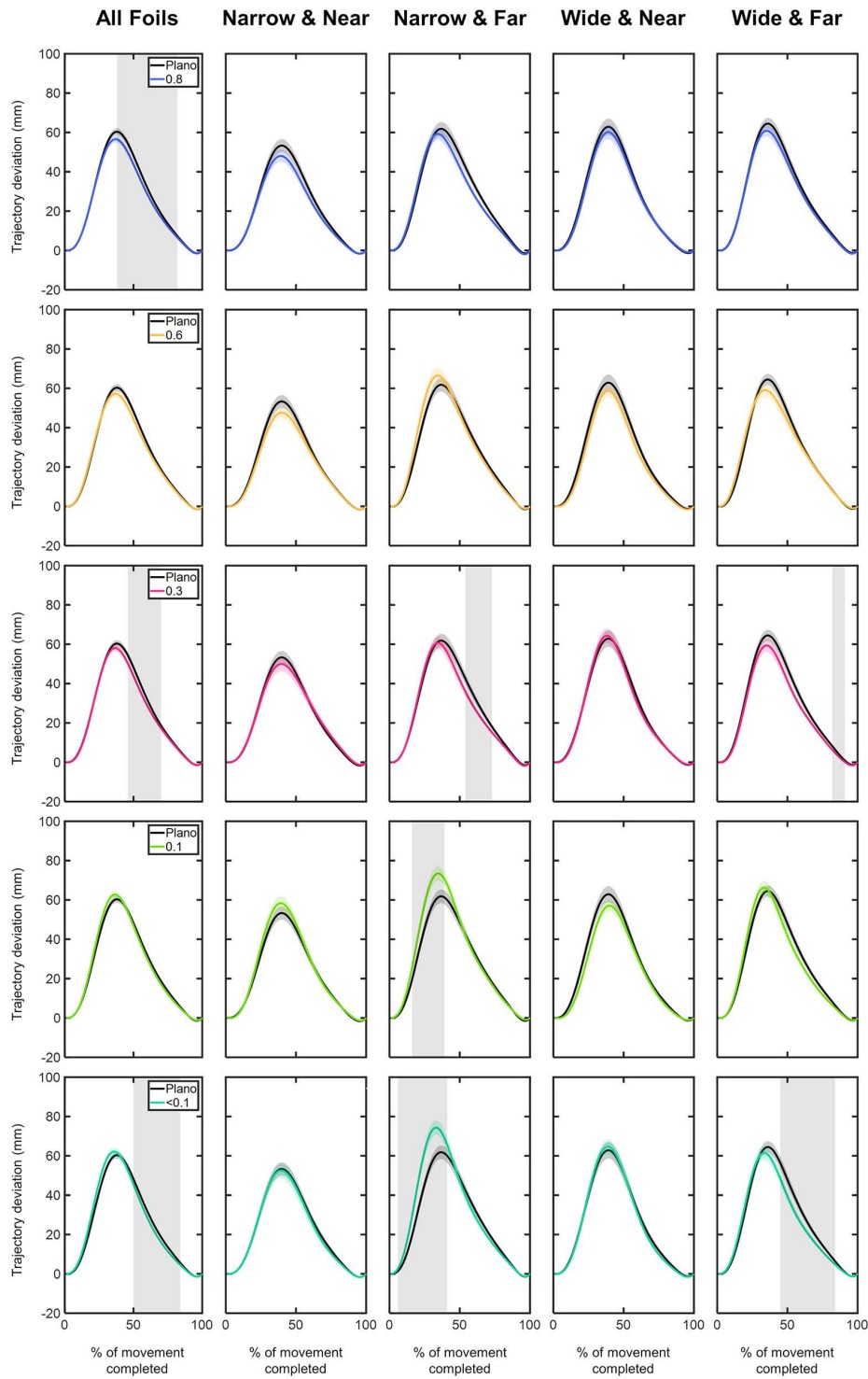

**Fig 12. Comparison of the z-component deviations (in mm) of the middle (MID) marker trajectory between the Plano condition and each Bangerter foil across different experimental conditions (object size and object distance).** Trajectory deviation values are plotted as a function of the percentage of movement completed, ranging from 0% (movement onset) to 100% (movement end). The values are represented by the mean of all trials from all participants, with shaded areas indicating the standard error of the mean. Grey shaded areas indicate statistically significant differences (FDR-corrected p values < 0.05) in the comparison.

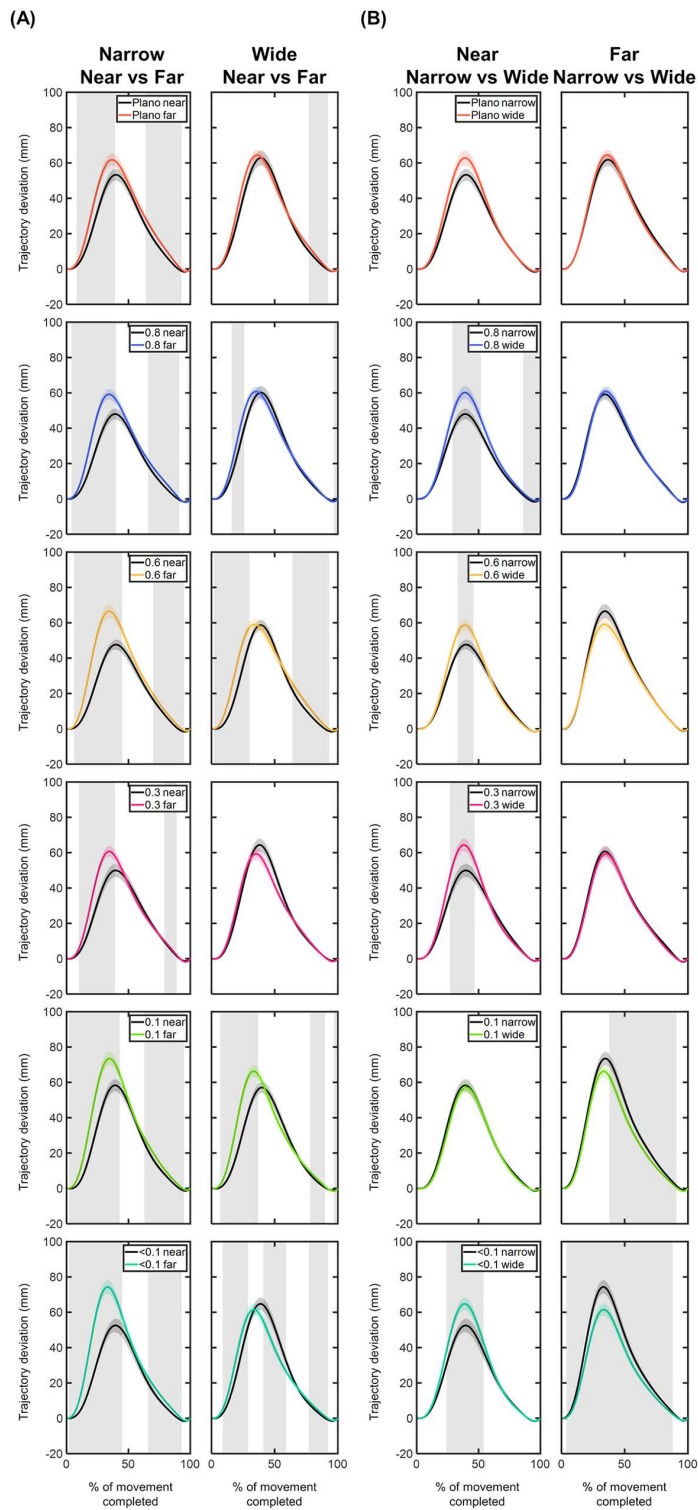

**Fig 13. Comparison of the z-component deviations (in mm) of the middle (MID) marker trajectory within each visual condition based on (A) object distance and (B) object size.** Trajectory deviation values are plotted as a function of the percentage of movement completed, ranging from 0% (movement onset) to 100% (movement end). The values are represented by the mean of all trials from all participants, with shaded areas indicating the standard error of the mean. Grey shaded areas indicate statistically significant differences (FDR-corrected p values < 0.05) in the comparison.

other studies, we found that the degree of visual degradation produced was not consistent with the degree of degradation predicted by the manufacturer [48–50]. Inconsistencies in the manufacturing process, non-uniform foil density, or the increased depth of field in real eyes are some of the reasons that may explain why Bangerter occlusion foils do not consistently reduce visual function as labeled [48–50].

Additionally, it is important to highlight that our outcomes showed discrepancies between the 0.8 and 0.6 foils in VA values, but not in CS values. A valuable explanation for this discrepancy can be found in the MTF analysis at different spatial frequencies (Fig 2A). As explained in the methods section, the visual tests for VA and CS have standardized but different methodological configurations. VA was evaluated using high-contrast black optotypes on a white background with variable sizes, while CS was assessed using optotypes on a gray background with fixed size and variable contrast. These configurations result in different spatial frequency spectra for VA and CS optotypes [51].

For example, regarding the CS test, the optotype comprises a mixture of low to medium spatial frequencies [38]. As shown in Fig 2A, the 0.8 and 0.6 foils presented similar MTF values within this frequency range. Such similarity in contrast transfer within the low to medium spatial frequencies could explain why both foils showed similar CS values.

Concerning the VA test, the spatial frequency of the optotype increases as the size decreases. For example, an optotype with a minimum angle of resolution of 1 minute of arc (equivalent to 1.0 on the decimal scale and 0.0 logMAR) corresponds to a spatial frequency of approximately 30 cpd [52,53]. In the current study, mean VA values of 0.08 and 0.20 logMAR were observed under the 0.8 and 0.6 foils, respectively, corresponding to spatial frequencies of approximately 19 and 25 cpd. As displayed in Fig 2A, within this spatial frequency range, the 0.8 foil presented lower MTF values compared to the 0.6 foil. This reduction in optical quality and contrast transfer capability in the high-frequency range may explain why worse VA values were observed under the 0.8 foil compared to the 0.6 foil.

These findings suggest that characterizing the ability to transfer contrast at different spatial frequencies is an important aspect to consider when analyzing the effects of occlusion foils on visual function.

## Reach dynamics

Overall, the visual function degradation negatively impacted the transport component. The observed effects were dependent on the level of foil density, kinematic phases of the movement, and intrinsic (size) and extrinsic (spatial location) properties of the object.

On average, the most significant effects were found with a reduction of 70% in VA and 55% in CS. These reductions in visual function led to longer MD, reduced AV, lower PV and PA, earlier appearance of PV, PA, and PD, and higher DecT and tLV (see Fig 4, S1–S3 and S5 Figs, and S1 Table).

Interestingly, the time course analysis of the MID marker's tangential velocity indicated that significant alterations in the transport component may arise even with slight to moderate visual degradation. For example, as shown in Fig 4, compared to non-degraded vision, under the 0.6, 0.1, and <0.1 foils, participants significantly reduced the velocity of the transport component during 22%, 50%, and 42% of the total movement duration, respectively. This appears to indicate that: (1) more severe visual function degradation impacted a larger portion of the movement; and (2) even reductions of 30% in VA and 15% in CS led participants to alter their hand's kinematics. The visual function degradation effects in the transport component predominantly occurred during the deceleration phase of the movement (starting from 37% of the completed movement). After the PV and under non-degraded vision, participants moved their hands towards the object at the highest velocity. In contrast, the acceleration phase, from MO to PV, was minimally influenced by the quality of visual feedback (Fig 4).

The interaction among object size, object distance, and foil density underlined the relevance of object properties when interpreting the impact of vision degradation on the transport component. On one hand, intrinsic and extrinsic object properties influenced the visual function degradation effects. During the acceleration phase, the effects of visual function degradation were more pronounced when participants performed the task with the wide object. Specifically, under the wide

object in both spatial locations, a reduction of 70% in VA and 55% in CS (foils 0.1 and <0.1) resulted in an average absolute percentage change of 7.8% in the kinematic variables related to the acceleration phase. In contrast, when manipulating the narrow object, these foils showed an absolute percentage change of 4.8% (see S2–S4 Figs). In the deceleration phase, the effects of visual function degradation were more evident when participants performed the task with the narrow object. Under the narrow object in both spatial locations, a reduction of 70% in VA and 55% in CS (foils 0.1 and <0.1) led to a mean absolute percentage change of 7.0% in the kinematic variables related to the deceleration phase. When manipulating the wide object, these foils showed a mean absolute percentage change of 3.4% (see S2–S5 Figs). These findings appear to indicate that the effects of visual function degradation were approximately 1.6 times greater with the wide object during the acceleration phase. In contrast, during the deceleration phase, the narrow object intensified the visual degradation effects by a factor of approximately 2 compared to the wide object.

On the other hand, the level of degradation of visual function influenced the relationship between object properties (size and distance) and the transport component. Scientific literature has widely analyzed the effects of varying object size and distance on the transport component. Robust conclusions have been reported about the effects object distance: reaching dynamics (e.g., movement duration or wrist velocity) increase with distance [54–58]. Our findings agree: transport component variables (MD, AV, PV, PA, PD, DecT, tLV, etc. see S1–S5 Figs) increased with object distance. Moreover, our results remained consistent for both sizes and under all visual conditions (Fig 5A). Interestingly, the effects of object distance on the transport component occurred on average up to 70% of the completed movement. In movement intervals after PV, both distances resulted in similar kinematic values (Fig 5A).

With respect to object size, some controversy is offered by the scientific literature. Some authors report no effect on the transport component [57,59], while others describe an increase of movement duration and deceleration phase with decreasing size [54,56,60,61]. A smaller contact surface given by a smaller object size has been reported as a possible interpretation for these results [62]. Our outcomes showed that, on average, under all visual conditions and both distances, variables such as AV, PV, or PA did not change with size (S2 Fig). However, variables such as MD, DecT, or tLV were higher with increasing size. At near distance, the wide size led to an increase in MD, DecT or tLV as long as the reduction in VA and CS was less than 50% and 30%, respectively (visual conditions: from Plano to 0.3 foil – see S1 and S5 Figs). In contrast, at far distance, the narrow size resulted in an increase in MD, DecT or tLV as long as the reduction in VA and CS was greater than 80% and 65%, respectively (visual condition: <0.1 foil – see S1 and S5 Figs). As noted, the effects appeared primarily in the deceleration phase reflecting the online motor control of prehension actions. Upon movement onset, the sensorimotor system makes necessary adjustments based on the received visual feedback to ensure controlled and precise reaching [63,64]. In the current experiment, the larger contact surface of the wide object at a near distance, combined with appropriate visual function, allowed participants to make more controlled sensorimotor adjustments, thereby increasing the deceleration phase. In contrast, the smaller contact surface of the narrow object at a far distance, along with highly degraded visual function, increased the need for maximum motor precision, thus lengthening the deceleration phase.

In short, our findings seem to suggest that intrinsic information derived from object's size is connected to the transport component, and this relationship may be modulated by the quality of visual feedback and extrinsic information from the object's distance.

## Grasp dynamics

During the prehension task, participants progressively increased their grip aperture, reaching maximum values at approximately 60% of the total movement duration. They then adjusted their grip aperture based on the object's size until achieving a successful grasp. This pattern remained consistent, although influenced by the induced visual function degradation. The effects of visual degradation on the manipulation component were dependent on foil density, and object's size and distance.

AGA, and PGA were affected by visual degradation. Participants over-scaled their grip aperture when performing the task under those foil densities that degraded their VA and CS by more than 70% and 55%, respectively. Vision degradation also influenced the timing of the manipulation component. Higher levels of visual degradation were associated with increased ttPGA and tPGA-OLO values and decreased % PGA values. On average, the 0.1 and <0.1 foils caused participants to require approximately 30 ms more for both ttPGA and tPGA-OLO and shifted the PGA to occur approximately 2% earlier.

The time course analysis of grip aperture supported these findings (Fig 6). First, 0.1 and <0.1 foils triggered over-scaling, particularly when participants manipulated the narrow object located at far (Fig 6). Second, over-scaling primarily occurred within the second third of the completed movement. After PGA, grip aperture values did not appear to be influenced by the quality of visual feedback.

The interaction among object size, object distance, and foil density also highlighted the importance of object properties when assessing the impact of vision degradation on the manipulation component. The effects of visual function degradation were approximately twice as high when participants performed the task with the narrow object. At both distances, a reduction of 70% in VA and 55% in CS (foils 0.1 and <0.1) resulted in an average absolute percentage change of 5.5% in the manipulation component variables for the narrow size. In contrast, when manipulating the wide object, these foils showed an absolute percentage change of 2.8% (S6 and S7 Figs).

Additionally, the level of visual degradation influenced the relationship between object size and distance, and the manipulation component. In agreement with other studies [54,59,65,66], grip aperture was adjusted proportionally to the object's size: larger objects resulted in larger apertures. Interestingly, the differences in grip aperture between the two object sizes became statistically significant starting from approximately 15% of the movement completed. This seemed to be influenced by the visual feedback quality, as the observed differences between both object sizes appeared later in time when visual function was more degraded. On average, under stronger foil densities (0.1 and <0.1), the differences emerged at approximately 27% of the completed movement. In contrast, under lighter densities, differences appeared at approximately 20% of the completed movement (Fig 7B).

The scientific literature provides conflicting evidence regarding the effects of object distance on the manipulation component. While some authors report no effects [54,59], others reveal an increase in PGA with increasing distance [55,56]. Our results showed that greater distances lead to higher values in AGA, % PGA, ttPGA, tPGA-OLO, but not in PGA. These results were more consistent with the narrow size at both distances (S6 and S7 Figs). The time course of grip aperture indicated that participants maintained higher grip aperture values for the far-positioned object in phases both before and after the PGA (Fig 7A). In fact, under the narrow size, a reduction of 50% or more in VA and 30% or more in CS led to these differences occurring on average in 80% of the total movement duration. Under the strongest foil density (<0.1 foil), participants maintained higher grip aperture values for far-positioned object for about 90% of the total movement duration (Fig 7A). As mentioned, our results showed no effect of object distance on PGA, except under the highest visual function degradation (S6 Fig). Under the <0.1 foil, with reductions of more than 80% in VA and 65% in CS, average PGA values were higher at far distance when participants manipulated the narrow size. However, surprisingly, with the wide size, under the <0.1 foil, PGA values were higher at near distance (S6 Fig). Indeed, these results were strongly supported by the findings from the time course analysis of grip aperture (Fig 7A). This seems to indicate that with the narrow object at far and the wide object at near, along with highly degraded visual feedback, the manipulation component over-scaled the PGA to provide a greater safety margin and ensure a successful grasp of the object.

Our results suggest that the manipulation component is connected to the visual information derived from the object's size and distance, with this relationship being adjusted according to the visual feedback quality.

## Spatial kinematics

Visual function degradation altered the x, y, and z components of the MID marker trajectory during the execution of the reach-to-grasp task. The impact relied on the foil density level, movement phase, and object's size and distance.

Overall, reductions greater than 70% in VA and greater than 55% in CS resulted in larger horizontal deviations, greater forward displacements, and higher elevation. These findings were corroborated both in the time course of the x, y, and z components (Figs 8, 10, 12) and in the mean values of PL, MLD, and MVD (S8 Fig and S1 Table).

The time course analysis revealed three main aspects. First, the largest horizontal deviations occurred under greater visual degradations, during the first half of the movement when participants manipulated the narrow object located at far (Fig 8). Second, alterations in the y and z components appeared even with moderate visual function degradations, starting at 30% in VA and 15% in CS. These effects were more pronounced when manipulating the narrow-sized object located at far (Figs 10 and 12). Third, the degradation of visual function primarily affected the trajectory of the z component starting from the MVD point, within the second half of the movement. However, this effect was not consistent, as it was influenced by the level of visual degradation, as well as the size and distance of the object. For example, alterations in the z component appeared during the first half of the movement under conditions of higher degradation, smaller size, and larger distance (Fig 12).

The interaction between the object's size and distance, and foil density emphasized the relevance of object properties when evaluating the effects of vision degradation on spatial components of the MID marker trajectory. The effects of visual function degradation were twice as high when participants performed the task with the narrow object. At both distances, a reduction of 70% in VA and 55% in CS (foils 0.1 and <0.1) resulted in an average absolute percentage change of 8% in MLD and MVD variables. When manipulating the wide object, these foils showed an absolute percentage change of 4% (S8 Fig). With the narrow object at far distance, the effects of visual function degradation were even more prominent: approximately three times greater compared to the wide size. Foils 0.1 and <0.1 generated an average absolute percentage change of 12.4% in MLD and 10.8% in MVD. For the wide size at far, the mean absolute percentage changes were 4.1% in MLD and 3.4% in MVD.

The level of degradation of visual function influenced the relationship between object properties (distance and size) and spatial components. Regarding the object's distance, deviations in the x, y, and z components increased with distance. In the x component, the largest deviations were observed with the narrow object under visual degradations greater than 70% in VA and 55% in CS (Fig 9A). Differences in deviations between both distances appeared for approximately 90% of the total movement duration. In the y component, the largest deviations remained constant across all visual conditions and both sizes (Fig 11A). In the z component, deviations appeared for approximately 65% of the total movement duration, within both the first and second halves of the movement. Differences between both distances emerged under all visual conditions, being more evident with the narrow object and more degraded visual conditions (Fig 13A; 0.1 and <0.1 foils).

With respect to object size, deviations in the x, y, and z components generally increased with size. In the x component, the largest deviations caused by the wide size occurred mainly under visual degradations greater than 70% in VA and 55% in CS when the object was at near distance (Fig 9B). In the z component, at near, the largest deviations generated by the wide size occurred mainly within the first half of the movement. Surprisingly, at far, the 0.1 and <0.1 foils led to the opposite. Visual degradations exceeding 70% in VA and 55% in CS resulted in larger deviations in the z component for narrow size compared to the wide size (Fig 13B).

All these adjustments seen in the spatial components may be an attempt by participants to minimize variability and errors during transport, increase the probability of object contact, and ensure successful object manipulation. Our findings underlined the importance of the visual feedback quality in the online correction of spatial components.

## Final remarks

The results of this study demonstrate that the degradation of visual function in normal-sighted individuals directly altered the reach and grasp components of prehension movements.

Participants exhibited longer movement durations, velocity profiles with reduced single-peaked bell shapes, lower acceleration peaks, slower and more prolonged deceleration phases, over-scaled hand grip apertures, and stereotyped

curved three-dimensional trajectories with larger deviation and path length. The three-dimensional kinematic analysis showed that these effects relied on the level of visual function degradation applied.

Population-based epidemiological studies and the World Health Organization define near vision impairment as having near VA at 40 cm less than N6 or M0.8, even with the best correction for distance VA [67,68]. These values are equivalent to 0.5 on the decimal scale and 0.3 logMAR. In our study, we considered VA under the control condition (Plano) as 100% vision, with participants achieving average values of 1.25 on the decimal scale (−0.10 logMAR). Therefore, a VA of 0.5 on the decimal scale would represent a 60% reduction in VA. As previously observed, a reduction in near VA greater than 60% was found under 0.1 and <0.1 foils. Although the most significant effects on kinematic measures were observed with reductions exceeding 70% in VA and 55% in CS, our results indicate that even visual function degradations starting at 30% in VA and 15% in CS (foils 0.8 and 0.6; VA ≤ 0.83 on the decimal scale) led participants to adopt compensatory strategies during the reach-to-grasp action.

The levels of visual function degradation at which we began to observe changes in the kinematic analysis are consistent with those reported by other authors on manual tracing tasks [69]. Domkin and colleagues categorized participants into four visual groups based on their best corrected binocular VA. Each group then completed manual tracing tasks on a screen tablet, which included five tracing patterns of varying difficulty. As expected, tracing errors increased with the complexity of the pattern and the reduction in VA. Notably, the authors found that performance in these manual tracing tasks began to decline when VA dropped below the 0.11–0.27 logMAR range (0.78–0.54 on the decimal scale). Similarly, in our study, within the 0.08–0.20 logMAR range (0.83–0.63 on the decimal scale of monocular VA), the first compensatory aftereffects on the transport, and spatial components of the reach-to-grasp movement began to appear. In Domkin and colleagues' study, the deterioration of manual motor actions was defined by tracing errors, evaluated based on the spatial relationships between the reference stimulus and the manual tracing [69]. In our study, we refer to compensatory mechanisms rather than errors. These mechanisms represent variations in kinematic components due to visual degradation when compared to the control condition. Our results align with the idea that the kinematic changes were a consequence of a motor strategy used to cope with the limitation imposed on visual feedback [27,28]. These compensatory mechanisms were likely applied to enhance confidence in the transport component, the probability of object contact, and the accuracy in manipulation [28,40]. Overall, our findings demonstrate that even subtle reductions in visual function can produce measurable aftereffects on hand motor performance.

The occlusion foils used in the current study are widely recognized for their ability to attenuate high spatial frequencies and degrade vision. This has led to their use in various clinical and scientific applications, such as treating visual conditions like amblyopia and diplopia [70,71] and simulating visual impairments [72–75]. Indeed, many studies have used this methodological approach, among others, to simulate visual suppression in individuals with normal vision to evaluate how vision degradation affects visuomotor coordination and dexterity in a variety of hand-based motor tasks, such as water-pouring, bead threading, pegboard, and driving [36,76–83]. For example, Sheppard and colleagues (2021) found that reduced VA and CS negatively affected the performance of sensorimotor tasks. This impact was evident regardless of whether the vision impairment was unilateral or bilateral, leading to decreased speed and precision in task execution [82]. Melmoth et al. (2007) examined the effects of altered vergence and binocular disparity cues on reach-to-grasp movements. To achieve this, the authors placed plus lenses and prims over the visual fields of the participants. The study demonstrated that such visual cues play distinct roles in both the planning and execution of the movement and are essential for selecting the correct endpoint position [78]. Similar studies can also be found in patients with visual impairments. Pardhan's work examined the motor performance of reaching tasks in individuals with central visual impairments and found that their ability to perceive and reach objects was significantly compromised. These participants required longer times to initiate movements and spent more time after reaching maximum grip aperture values, indicating the need for extended periods to perform grasping tasks accurately [33]. Another study by the same author explored motor performance as a function of the affected retinal area. This study compared the motion performance of participants with central

field loss due to age-related macular degeneration and those with peripheral visual field loss caused by glaucoma. The results showed that participants with central visual field loss performed worse when completing the task. These findings were associated with a loss of VA and CS, highlighting the importance of central visual function in prehension movements [45]. Similarly, other studies have evaluated the effects of real or simulated binocular stereo vision losses in children and adults during reach-to-grasp activities [79,83]. These studies have reported longer movement durations, increased reach-to-grasp errors, among other effects, highlighting the importance of binocular stereo vision for skilled prehension movements. Despite the methodological differences, all in all, these studies highlight two key aspects: the functional impact of degraded vision on these activities and the need to objectively quantify the relationship between visual function and visuomotor performance. Despite evident methodological differences, our study addresses both aspects. Our findings confirm that visual degradation significantly impacts prehension performance. Additionally, they provide an objective and detailed quantification of how, when, and where these effects manifest in motor skills. To our knowledge, this is the first study to offer such a rigorous objective description of the impact of different levels of visual function degradation on the components of manual prehension. Here, we focused on analyzing the temporal aspects of reach and grasp dynamics, as well as spatial kinematics. Our methodological approach allowed us to detect with high accuracy subtle changes in the three-dimensional hand data, providing a detailed and comprehensive view of how variations in visual function affect manual prehension performance.

### Limitations of the study

It is important to note that the visual function degradation was induced in a group of young individuals with normal vision. This means we are not dealing with natural visual degradation, as we are excluding the visual adaptation processes. Consequently, the visual degradation and its aftereffects on manual prehension performance observed in our results may be more pronounced than those seen in individuals with real and different levels of visual function.

Another important aspect to consider is the number of repetitions used to test each condition. While we acknowledge that a larger number of repetitions would provide higher statistical power and potentially more robust results, the experimental design of this study aimed to balance the need for sufficient repetitions with minimizing noisy data due to loss of concentration and fatigue, as well as reducing learning effects. To provide a clearer understanding of the effect sizes obtained, partial eta squared, and Cohen's f values have been included (S1 Table).

Furthermore, the use of monocular viewing conditions should be also considered. Although binocular vision enhances prehension performance, it also introduces complex neural processes that could confound our expected results. Monocular viewing was chosen to maintain controlled and standardized experimental conditions. We acknowledge this limitation and its impact on the interpretation of our findings.

Despite these limitations, our results align with the scientific literature and can therefore be used as a reference for future research and clinical applications aimed at evaluating the impact of vision on visuomotor skills in daily activities.

### Conclusion

In conclusion, our data provide new evidence on the effects of visual function degradation on the kinematics of manual prehension movements. Our findings suggest that participants adjusted their transport, manipulation, and spatial components based on the level of visual function degradation. The most significant effects on motor performance were observed with reductions of 70% in VA and 55% in CS. However, even subtle reductions greater than 30% in VA and 15% in CS were sufficient to generate noticeable changes in the kinematic variables analyzed. As a result, visual function degradation led participants to adopt more conservative and controlled movement strategies to minimize errors and ensure a successful grasp of the object. This study demonstrates the importance of objective kinematic analysis in evaluating the effects of visual function on motor skills. Our results are particularly relevant for individuals with vision disorders and can

be helpful in developing training strategies for individuals with visual function loss, offering a solid foundation for future research and clinical applications.

## Supporting information

**S1 Fig. General Kinematics: Mean values for Movement Duration across all visual conditions: Plano, 0.8, 0.6, 0.3, 0.1, and <0.1, based on the object's size and distance.** Errors represent the standard error of the mean.
(TIF)

**S2 Fig. Reach Dynamics I: Mean values for Average Velocity, Peak Velocity, Peak Acceleration, and Peak Deceleration across all visual conditions: Plano, 0.8, 0.6, 0.3, 0.1, and <0.1, based on the object's size and distance.** Errors represent the standard error of the mean.
(TIF)

**S3 Fig. Reach Dynamics II: Mean values for Percentage Peak Velocity, Percentage Peak Acceleration, and Percentage Peak Deceleration across all visual conditions: Plano, 0.8, 0.6, 0.3, 0.1, and <0.1, based on the object's size and distance.** Errors represent the standard error of the mean.
(TIF)

**S4 Fig. Reach Dynamics III: Mean values for Time to Peak Velocity, Time to Peak Acceleration, Time to Peak Deceleration, and Time from Peak Deceleration to Object Initial Contact across all visual conditions: Plano, 0.8, 0.6, 0.3, 0.1, and <0.1, based on the object's size and distance.** Errors represent the standard error of the mean.
(TIF)

**S5 Fig. Reach Dynamics IV: Mean values for Deceleration Time, Normalized Deceleration Time, Time spent at Low Velocity, and Normalized Time spent at Low Velocity across all visual conditions: Plano, 0.8, 0.6, 0.3, 0.1, and <0.1, based on the object's size and distance.** Errors represent the standard error of the mean.
(TIF)

**S6 Fig. Grasp Dynamics I: Mean values for Average Grip Aperture, Peak Grip Aperture, and Percentage Peak Grip Aperture across all visual conditions: Plano, 0.8, 0.6, 0.3, 0.1, and <0.1, based on the object's size and distance.** Errors represent the standard error of the mean.
(TIF)

**S7 Fig. Grasp Dynamics II: Mean values for Time to Peak Grip Aperture, Time from Peak Grip Aperture to Object Lift Onset, and Time Post Contact across all visual conditions: Plano, 0.8, 0.6, 0.3, 0.1, and <0.1, based on the object's size and distance.** Errors represent the standard error of the mean.
(TIF)

**S8 Fig. Spatial Kinematics: Mean values for Path Length, Maximum Lateral Deviation, and Maximum Vertical Deviation across all visual conditions: Plano, 0.8, 0.6, 0.3, 0.1, and <0.1, based on the object's size and distance.** Errors represent the standard error of the mean.
(TIF)

**S1 Table. Statistical results for each kinematic parameter from the three-way repeated measures ANOVA analysis.** The columns are defined as follows: "df" represents the degrees of freedom for the factor, "df (error)" for the error term, "F" is the F-value, "P-value" indicates the significance level, "Partial Eta Squared" measures the effect size, and "Cohen's f" indicates the strength of the relationship between the factor and the dependent variable. The columns labeled

as "F", "S", and "D", stand for the following experimental aspects: visual condition, object size, and object distance, respectively. Interactions between these factors are labeled as "FS", "FD", "SD", and "F*S*D".
(DOCX)

## Author contributions

**Conceptualization:** Pablo Sanz Diez, Sandra Gisbert.

**Formal analysis:** Pablo Sanz Diez, Sandra Gisbert.

**Funding acquisition:** Annalisa Bosco, Patrizia Fattori, Siegfried Wahl.

**Investigation:** Pablo Sanz Diez, Sandra Gisbert.

**Methodology:** Pablo Sanz Diez, Sandra Gisbert.

**Resources:** Annalisa Bosco, Augusto Arias, Patrizia Fattori, Siegfried Wahl.

**Software:** Pablo Sanz Diez, Sandra Gisbert, Annalisa Bosco, Augusto Arias.

**Supervision:** Annalisa Bosco, Patrizia Fattori, Siegfried Wahl.

**Writing – original draft:** Pablo Sanz Diez, Sandra Gisbert.

**Writing – review & editing:** Pablo Sanz Diez, Sandra Gisbert, Annalisa Bosco, Augusto Arias, Patrizia Fattori, Siegfried Wahl.

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
