## [Decision Letter · Decision Letter 0]

15 Dec 2024

PONE-D-24-50802

Exploring the impact of visual function degradation on manual prehension movements in normal-sighted individuals

PLOS ONE

Dear Dr. Sanz Diez,

Thank you for submitting your manuscript to PLOS ONE. After careful consideration, we feel that it has merit but does not fully meet PLOS ONE’s publication criteria as it currently stands. Therefore, we invite you to submit a revised version of the manuscript that addresses the points raised during the review process.

Two reviewers provide their detailed feedback below. They both identify that there are important limitations in the study that should be addressed before making a final decision. One of these refers to the data sampling both with respect to the sample size and the replications of each condition (three trials each). Moreover, several aspects related to the analyses should be clarified, including the calculation of trajectory deviation based on the index finger data only. Finally, both reviewers provide useful suggestions about the visualization of the graphics, which I encourage you to consider.

We look forward to receiving your revised manuscript.

Kind regards,

Dimitris Voudouris

Academic Editor

PLOS ONE

Journal Requirements: When submitting your revision, we need you to address these additional requirements. 1. Please ensure that your manuscript meets PLOS ONE's style requirements, including those for file naming. The PLOS ONE style templates can be found at https://journals.plos.org/plosone/s/file?id=wjVg/PLOSOne_formatting_sample_main_body.pdf and https://journals.plos.org/plosone/s/file?id=ba62/PLOSOne_formatting_sample_title_authors_affiliations.pdf 2. Please note that PLOS ONE has specific guidelines on code sharing for submissions in which author-generated code underpins the findings in the manuscript. In these cases, we expect all author-generated code to be made available without restrictions upon publication of the work. Please review our guidelines at https://journals.plos.org/plosone/s/materials-and-software-sharing#loc-sharing-code and ensure that your code is shared in a way that follows best practice and facilitates reproducibility and reuse. 3. Thank you for stating the following financial disclosure:  [This work was supported by the PLACES project which has received funding from the European Union’s Horizon 2020 research and innovation programme under the Marie Skodowska-Curie grant agreement No 101086206, and by the MAIA project which has received funding from the European Union’s Horizon 2020 research and innovation programme under the Marie Skodowska-Curie grant agreement No 951910.].  Please state what role the funders took in the study.  If the funders had no role, please state: ""The funders had no role in study design, data collection and analysis, decision to publish, or preparation of the manuscript."" If this statement is not correct you must amend it as needed. Please include this amended Role of Funder statement in your cover letter; we will change the online submission form on your behalf. 4. We note that your Data Availability Statement is currently as follows: [All relevant data are within the manuscript and its Supporting Information files.] Please confirm at this time whether or not your submission contains all raw data required to replicate the results of your study. Authors must share the “minimal data set” for their submission. PLOS defines the minimal data set to consist of the data required to replicate all study findings reported in the article, as well as related metadata and methods (https://journals.plos.org/plosone/s/data-availability#loc-minimal-data-set-definition). For example, authors should submit the following data: - The values behind the means, standard deviations and other measures reported;- The values used to build graphs;- The points extracted from images for analysis. Authors do not need to submit their entire data set if only a portion of the data was used in the reported study. If your submission does not contain these data, please either upload them as Supporting Information files or deposit them to a stable, public repository and provide us with the relevant URLs, DOIs, or accession numbers. For a list of recommended repositories, please see https://journals.plos.org/plosone/s/recommended-repositories. If there are ethical or legal restrictions on sharing a de-identified data set, please explain them in detail (e.g., data contain potentially sensitive information, data are owned by a third-party organization, etc.) and who has imposed them (e.g., an ethics committee). Please also provide contact information for a data access committee, ethics committee, or other institutional body to which data requests may be sent. If data are owned by a third party, please indicate how others may request data access.

Reviewers' comments:

Reviewer's Responses to Questions

**Comments to the Author**

1. Is the manuscript technically sound, and do the data support the conclusions?

Reviewer #1: Partly

2. Has the statistical analysis been performed appropriately and rigorously?

Reviewer #1: Yes

3. Have the authors made all data underlying the findings in their manuscript fully available?

Reviewer #1: No

4. Is the manuscript presented in an intelligible fashion and written in standard English?

Reviewer #1: Yes

5. Review Comments to the Author

Reviewer #1: RE Exploring the impact of visual function degradation on manual prehension movements in normal-sighted individuals

I really enjoyed reading this paper. The authors explore the impact of different levels of visual function degradation on prehension movements in young adults with normal vision. Degradation was found to alter numerous aspects of the movement and the authors claim that even subtle reductions in VA and CS triggered compensatory mechanisms. The paper is well-written, comprehensive and clear, and the figures are exceptionally helpful. I have a few suggestions for how it might be improved. My one main concern is that the number of trials is really small (see comment below).

Data availability

The authors state that all data are available without restriction but I couldn’t see a link.

Introduction:

Why have you gone for monocular viewing throughout the experiment? Binocular vision, with one eye or both blurred, would also have been an option so I’m interested in why you didn’t go for that and think you need to justify your choice in the Intro. Are your ruling our stereopsis complications?

Methods:

1. Would you consider including a table that has decimal, logMAR, and Snellen so readers can convert easily? I think in logMAR, so had to look up what the equivalent decimal score meant to understand the density of the filters and ended up writing myself a table to refer to. I appreciate that the actual logMAR score that the filters induce is not what they state they will, but presumably the same could be said for the decimal score (they don’t reduce the VA by the stated amount). You could edit Figure 3 instead (which I found incredibly useful) if that suits better – your X axis could include both stated decimal and logMAR on the filters. Your call – don’t want this to end up complicating things so just a suggestion.

2. Line 175 – 24 trials under each visual condition. I can’t follow this – did participants not just read as far as they could on each chart and you scored it using the smallest/faintest letter read? Perhaps this is my unfamiliarity with the tests used but 24 trials to ascertain VA and CS feels like a lot. Perhaps you mean letters/optotypes rather than trials?

3. Why right eye? Why not test for eye dominance and use the dominant eye (or argue for using the non-dominant one!) I appreciate that eye dominance might not be a particularly useful thing to measure and use, so this Q is more out of curiosity than anything.

4. Line 190 – “Once grasp it” typo

5. Line 197 – participants wore an opaque lens over the right eye between each trial. How was this removed? And when? And why did they have to close their eyes between each trial if they had opaque lenses over both?

6. Line 201 – participants did 72 trials across 6 visual conditions, 2 object sizes and 2 object distances. This is only 3 trials per trial type (as you state). Is this enough? It’s typical to need at least 8 trials to get a reliable mean for performance. I’m really concerned that a lot of what you’ve found might just be noise. Can you at least report effect sizes?

7. Table 1 is really useful, but I can’t see any mention of the wrist markers. I’m wondering why you used three, and why you used index rather than wrist for reach onset, peak velocity etc? Were none of the wrist markers actually analysed?

8. Line 262: I’m interested in how you explore deviations to trajectories. You use a straight line between start and end to denote a virtual straight traj and explore deviations from this. But deviations are expected surely, as nobody moves in a straight line? Why do this rather than something like zero crossings in acceleration?

Results:

1. I initially thought you might be overstating your findings in terms of the effects of subtly degraded vision on performance. Perhaps I’m mistaken, but the in Reach Dynamics section it appears that the only time you get significant effects of visual condition are when vision is the most degraded – so plano vs <0.1 or 0.1? For Grasp Dynamics (see point below) I don’t know where differences lie. It seems that differences appear with subtle degradation when you’re examining Time course information onwards (line 472). This is still really interesting, but the way the paper is currently written/ordered I started to think you hadn’t found much of interest as the first sections of the results point to the idea that vision has to be pretty bad to find an effect on performance. I think this needs to be made more explicit in your Discussion (and abstract). The order of sections in your Results section is correct, I just think you could flag this in your Discussion – that effects are there, they’re just not evident if you only analyse the data in a certain way (reach dynamics)…

2. In the Grasp Dynamics Section why aren’t you including pairwise comparisons to interrogate main effects as you did in the Reach Dynamics Section? At the moment I can’t tell whether vision has to be degraded to a certain extent to find an effect

Discussion:

1. Here is where I think you overstate things in your first para (see first comment on Results section also) when you state “Our results demonstrate that the visual function degradation significantly impacted the kinematic and spatial performance of prehension movements during the evaluated task. In general, participants took longer to reach and grasp the object when they experienced a degradation of their visual function. This aspect was supported by a combination of kinematic and spatial factors such as a reduction in velocity and acceleration, more time spent decelerating and at low velocity, and movement trajectory adjustments”. For at least part of this it’s only true when vision is seriously degraded…. Having said that….

2. Lines 796 to 805 are a great example of where you’re not over-stating things. Comparing everything to actual VA reduction %, and to WHO criteria, is really helpful. The point that your 0.1 and <0.1 conditions aren’t actually that strong is an important one, and perhaps could be made earlier and if so allow you to make some of the claims that at the moment feel a bit strong/over-stated. It made me realise that you’re perhaps not over-stating things after all and that these “strong blur” conditions aren’t actually that strong – it’s not as if people can’t see anything!

3. Your conclusion is compelling and measured.

Reviewer #2:

This study investigates a straightforward question: how do reductions in visual acuity and contrast sensitivity (induced by Bangerter occlusion foils) affect grasping movements in otherwise normally-sighted participants? The authors report alterations in both the reach and grasp components of prehension movements, more visible with the more higher reductions in visual function. The findings reported in this study are of interest, however, there are several elements in the presentation and discussion that could be improved.

Major points:

The study was conducted in monocular viewing conditions. What was the reason for this? It is known that grasping actions are heavily impacted by the lack of stereo vision. A rationale should be provided for this choice and the effect of lacking stereo vision should be discussed.

Graphical representation of the results: the data represented in Table 2 are too dense to convey any information. I suggest representing the data in figures consisting of multiple subplots each showing on the x axis the foil variable, on y axis its dependent variable, and four lines representing the four possible combinations of size and distance. In this way, it can be easily seen which variables are impacted by the introduction of the foils,  what foil level does the disruption of the visual function affect that particular dependent variable, and, most importantly, also it would allow to assess the practical significance of the disruption. Also, have you considered plotting on the x axis the actual values of VA and CS measured for each participant?

Results: the analyses should focus more on the effects of the occlusion foils. Many of the reported analyses are just obvious: the fingers open more for the large object, movement duration is longer for the far object, etc. This comment is linked to one of my comments below where I suggest to make it clearer which variables were expected to be impacted especially by the introduction of the different foils.

Data availability: The authors have declared the all data are fully available without restrictions and that all relevant data are within the manuscript and supporting information files, but neither of these statements is correct. There is no link to a data repository and not all data are available within the manuscript.

Figures quality: I don't know if it is an effect of the journal submission process, but figures have very low resolution making many labels very difficult to read.

Other points:

Introduction:

Lines 63-66: Can the authors add a citation that supports this affirmation, especially the motor function?

Lines 67-69: There are a few studies on the topic that are worth to mention, e.g.:

https://doi.org/10.1016/j.visres.2014.11.009

https://doi.org/10.1097/OPX.0b013e31824c1b89

https://doi.org/10.1111/j.1475-1313.2010.00819.x

Participants:

Line 87: please remove the reference to the Declaration of Helsinki. The latest version requires the studies to be preregistered, which is not the case for this study.

Lines 96-98: does this mean that some participants had their vision corrected for this study, but they had uncorrected vision in their everyday life? If so, it means that they were facing not only the effect of the occlusion foils, but also the effect of a corrected vision they were not used so. Please clarify.

Subjective characterization:

Line 169: units are reported in decimal scale in this section and in LogMar units in the results section. It would be good to add the LogMar units here as well.   

Lines 177-178: VA and CS values vary with distance, so how come they were measured at 40 cm, but objects in the grasping experiment were positioned at 25 and 50 cm?

Data processing and kinematic parameters:

Line 230: which noise removal process was used?

Line 233: unclear what you mean when you write that you "reconstructed unregistered motion". Please clarify.

Lines 229-236: no trials were eliminated during this process?

Line 247: the authors have analyzed 29 different dependent variables. Some of these variables are quite unconventional, for example, Average Velocity, Time spent at Low Velocity, Average Grip Aperture. Also, many of these variables can be highly correlated. What was the reason for this choice? It would be helpful to discuss what were the expected changes on these variables and by this try to group them according to their meaning. To be clear, I'm not suggesting that new indices should be created based on the existing variables, but just making it clearer which variables should be impacted especially by the introduction of the different foils.

Lines 262-270: were these deviations in x, y, z calculated on the basis of the temporally normalized trajectories?

Table 1: why do you define the exact same variable twice? I refer to the definitions of Object Lift Onset and Movement End.

Table 1: since you define the onset of the movement as Reach Onset, use this term consistently in the table and in the paper instead of introducing the term movement onset.

Subjective characterization of Bangerter foils:

Lines 313-315 and 317-318: The 0.6 foil should have reduced the VA and CS more than the 0.8 foil. Any other explanation on why this happened than what is currently proposed in the discussion? Do all participants show a larger impact of the 0.8 than the 0.6 foil? Could it be that the two foils have been accidentally swapped?

Table 2: I would suggest putting this table in the Supplementary material. Also because the information displayed is only partial: the values of each variable are only averages over size and distance for each foil, and the p-values are shown without their F values and degrees of freedom (which are then reported in the main text anyway).

Spatial kinematics: some of the variables are computed by taking the index finger as reference. However, this choice might not be optimal to investigate spatial kinematics, because the index finger trajectories are by definition affected by some of the properties of the objects that were changed in the experiment and/or by effects that have more to do with the changes in grip aperture than in the path taken by the hand to reach the object. For example, the trajectory of the index finger will always be longer for the larger than for the smaller object (the index has to travel a longer way to reach the back of the larger object). Or, the trajectory of the index finger will be by definition longer if a certain condition requires larger grip apertures. To solve this problem the authors should take as reference the average point (in 3D) between the thumb and the index finger. This variable will more clearly show if the hand takes a different path in the different conditions.

Figures 4 to 13: These figures could be useful as Supplementary material and improved versions (focusing mainly on the effects of foils should be used in the main text). It is really not relevant for the topic of this study to show that tangential velocity was higher for objects far away, or, that the grip aperture was larger for the larger objects. What any reader would like to see are the differences between visual conditions. For example, first column of Fig. 4, first column of Fig. 6, first column of Figs. 8, 10, 12 (with the average thumb-index finger trajectory instead of index only trajectory).

Discussion:

Some studies have found that the severity of visuomotor deficits was likely associated with the level of stereoacuity loss rather than the visual acuity loss or contrast sensitivity. Further discussion on the topic would strengthen the discussion.

Lines 825 to 834: It is possible that the 0.6 foil and 0.8 foil were swapped or misclassified? Another question arises from the dissonance between the visual function anomalies and kinematic analysis. If VA and CS are better in 0.8 what would be the reason for having worst performance under 0.6 foil? Based on the explanation of the authors, larger visual angle would lead the elements of the 0.6 foil to cover a larger area of the visual field. That would have influences on the VA or CS, which was not the case. Perhaps the contradictory findings may be related with the distances used for the tasks that were different from the distances used to measure VA and CS?

Lines 840 to 843: I think the discussion on previous studies is insufficient. It would be good to understand how the results of this study compare with previous research. For example, did the authors found similar results to previous studies using plus lenses (e.g., I: 10.1007/s00221-007-1041-x)? Although the authors cite the Melmoth study, a more detailed discussion should be added. It is also important to understand how the results compare with previous studies in patients with visual impairment.

Limitations of the study:

The fact that monocular vision has been used should be discussed here. And, also, the fact that sample of data is not really large, in terms of number of participants, but even more so in terms of sizes/distances used and repetitions for each combination (only three unique values for each participant).

6. PLOS authors have the option to publish the peer review history of their article (what does this mean? ). If published, this will include your full peer review and any attached files.

**Do you want your identity to be public for this peer review?** For information about this choice, including consent withdrawal, please see our Privacy Policy .

Reviewer #1: No

---

## [Author Response · Author response to Decision Letter 1]

2 Mar 2025

Thank you for giving us the opportunity to submit a revised version of the manuscript “Exploring the impact of visual function degradation on manual prehension movements in normal-sighted individuals”. We are grateful to you and the two Reviewers for taking the time and effort to assess the manuscript and for their helpful and constructive comments, which have brought valuable improvements to our manuscript.

We have revised our manuscript accordingly and respond to each of their points as described below. As mentioned in the Cover Letter, for some comments, we are currently conducting additional measurements to provide accurate responses. We appreciate your understanding and patience as we complete them.

The Reviewers’ comments are shown in black, our responses in green.

---

## [Editor Report · Decision Letter 1]

14 Mar 2025

We look forward to receiving your revised manuscript.

Kind regards,

Dimitris Voudouris

Academic Editor

PLOS ONE

Journal Requirements:

**Additional Editor Comments:**

Thank you for submitting your manuscript to PLOS ONE. We have noticed that some of your responses to the reviewers' comments are incomplete. We are returning the manuscript back to you so that you can fully address all raised comments before we invite the reviewers to evaluate the revised version of your submission.

---

## [Author Response · Author response to Decision Letter 2]

28 Apr 2025

Thank you for giving us the opportunity to submit a revised version of the manuscript "Exploring the impact of visual function degradation on manual prehension movements in normal-sighted individuals". We are grateful to you and the two Reviewers for taking the time and effort to assess the manuscript and for their helpful and constructive comments, which have brought valuable improvements to our manuscript. We have revised our manuscript accordingly and responded to each of their points in the "Response to Reviewers" file.

---

## [Decision Letter · Decision Letter 2]

30 May 2025

Dear Dr. Sanz Diez,

Thank you for submitting your manuscript to PLOS ONE. The reviewers are satisfied to a large extent with the revised version, yet there are some remaining comments that should be addressed. Please refer below to the detailed comments by Reviewer 2 for details. In short, I recommend to reduce emphasis on some known effects on grasping and rather highlight those related to the different levels of visual impairment. This relates also to a couple of figures, as they include data that remove emphasis from your main findings. I appreciate the transparency of reporting the detailed data, but you may want to consider bringing some of the figures to the supplement and keep figures showing the main effects (visual impairment) in the manuscript. One idea could be to show the timecourses for the different visual impairement levels, without including the effects of size or distance. Please refer to the comments below for more details and suggestions.

We look forward to receiving your revised manuscript.

Kind regards,

Dimitris Voudouris

Academic Editor

PLOS ONE

Journal Requirements:

Reviewers' comments:

Reviewer's Responses to Questions

**Comments to the Author**

Reviewer #1: All comments have been addressed

Reviewer #2: (No Response)

2. Is the manuscript technically sound, and do the data support the conclusions?

Reviewer #1: (No Response)

Reviewer #2: Partly

3. Has the statistical analysis been performed appropriately and rigorously?

Reviewer #1: (No Response)

Reviewer #2: Yes

4. Have the authors made all data underlying the findings in their manuscript fully available?

Reviewer #1: (No Response)

Reviewer #2: No

5. Is the manuscript presented in an intelligible fashion and written in standard English?

Reviewer #1: (No Response)

Reviewer #2: Yes

Reviewer #1: Thanks for responding so comprehensively to all my comments and requests. I'm happy with all the responses and edits you have made.

Reviewer #2: - Clarifying which variables were expected to be impacted by the introduction of the different foils. The authors asked for further details about my comment from the first round of reviews, so I'll try to do my best. What I meant is that there are a number of effects that are already known in the literature (the effect of distance on velocity, duration, etc., the effect of size on various aspects of grip aperture), so these effects should not take the center stage in the manuscript. What should instead take the center stage is how the different levels of visual impairment affect the various measures. And, for some of these measures, there are some expected effects that the authors have (I imagine) hypothesized to happen. For example, the lower is the quality of vision, the longer and slower the movements should be with larger grip apertures. This type of hypotheses could be added at the beginning of the results section and then directly addressed by showing if they were actually supported by the data or not. Then, it is fine to mention that most of the study is actually very exploratory.

- This comment is also linked to the previous comment. Figures 4 to 13 are overwhelming, there are almost 250 subfigures in total. The authors should help direct the attention of the readers to the important aspects in their data (hopefully, steered by the hypothesis). Again, many panels show just obvious facts, for example, the fingers open more for the wide than the narrow object. The authors could reduce the number of figures/panels by focusing on the key findings relevant to the research question and move the other figures/panels to the supplementary material. However, if the authors and the editor are fine keeping this format, I'm not going to argue for further changes.

- Data availability: I appreciate that the authors have now added some data as supplementary material, but these are only presented as mean + sem. It would be helpful for the field to provide data of individual trials and participants.

- Statistics question: In the analyses about the temporal evolution of variables, paired t-tests were used for each step of the movement (101 tests for each of the almost 250 panels). It would be appropriate to apply some correction for multiple comparisons also in these cases, at least, to take into account the multiple tests that are performed within each panel.

**Do you want your identity to be public for this peer review?** For information about this choice, including consent withdrawal, please see our Privacy Policy

Reviewer #1: No

Reviewer #2: No

---

## [Author Response · Author response to Decision Letter 3]

14 Jul 2025

Response to Reviewers

We thank the Editor and Reviewers for their time and constructive feedback. We have carefully revised the manuscript in response to their comments, which we address point by point below. The Reviewers’ comments are shown in black, and our responses are in green.

Reviewer #1

Thanks for responding so comprehensively to all my comments and requests. I'm happy with all the responses and edits you have made.

Thank you for your positive comment. We appreciate your time and effort in reviewing our manuscript.

Reviewer #2

Clarifying which variables were expected to be impacted by the introduction of the different foils. The authors asked for further details about my comment from the first round of reviews, so I'll try to do my best. What I meant is that there are a number of effects that are already known in the literature (the effect of distance on velocity, duration, etc., the effect of size on various aspects of grip aperture), so these effects should not take the center stage in the manuscript. What should instead take the center stage is how the different levels of visual impairment affect the various measures. And, for some of these measures, there are some expected effects that the authors have (I imagine) hypothesized to happen. For example, the lower is the quality of vision, the longer and slower the movements should be with larger grip apertures. This type of hypotheses could be added at the beginning of the results section and then directly addressed by showing if they were actually supported by the data or not. Then, it is fine to mention that most of the study is actually very exploratory.

Thank you very much for your helpful clarification. We agree that stating the hypothesis is important to guide the reader’s understanding. Therefore, we have included our hypothesis at the end of the Introduction section to provide a concise overview of the expected effects.

We believe that including a highly detailed hypothesis addressing all possible variables involved could potentially confuse readers. Presenting a general hypothesis allows us to maintain clarity, helping readers interpret the expected results within a broad context.

This comment is also linked to the previous comment. Figures 4 to 13 are overwhelming, there are almost 250 subfigures in total. The authors should help direct the attention of the readers to the important aspects in their data (hopefully, steered by the hypothesis). Again, many panels show just obvious facts, for example, the fingers open more for the wide than the narrow object. The authors could reduce the number of figures/panels by focusing on the key findings relevant to the research question and move the other figures/panels to the supplementary material. However, if the authors and the editor are fine keeping this format, I'm not going to argue for further changes.

Thank you for your suggestion. We understand the Reviewer’s concern about the large number of plots included, however we do not believe this negatively affects manuscript comprehension. As mentioned in the first revision, the primary goal of our experiment was to explore how different levels of visual function degradation influence the reach and grasp components of manual prehension movements. While the visual factor is indeed central, we found it essential to consider interactions with object size and distance to provide the reader with a comprehensive understanding.

From our point of view, reorganizing or moving the figures to supplementary material could hinder rather than help the interpretation of results. Unless the Reviewer or the Editor feel otherwise, we would prefer to keep the current format.

Data availability: I appreciate that the authors have now added some data as supplementary material, but these are only presented as mean + sem. It would be helpful for the field to provide data of individual trials and participants.

Thank you for your suggestion. We agree, and therefore, we have uploaded the datasets to the Open Science Framework (OSF), including individual trial-level data for each participant.

The data is publicly available at: https://osf.io/wrja5/

Statistics question: In the analyses about the temporal evolution of variables, paired t-tests were used for each step of the movement (101 tests for each of the almost 250 panels). It would be appropriate to apply some correction for multiple comparisons also in these cases, at least, to take into account the multiple tests that are performed within each panel.

Thank you for bringing this important point to our attention. We acknowledge that this aspect was initially overlooked. We fully agree that applying a correction for multiple comparisons is essential to control the type I error rate. Accordingly, we have applied the Benjamini-Hochberg procedure to control the false discovery rate at an alpha level of < 0.05. All related results, figures, and statistical information have been updated in the revised version of the manuscript.

---

## [Decision Letter · Decision Letter 3]

29 Jul 2025

Exploring the impact of visual function degradation on manual prehension movements in normal-sighted individuals

PONE-D-24-50802R3

Dear Dr. Sanz Diez,

We’re pleased to inform you that your manuscript has been judged scientifically suitable for publication and will be formally accepted for publication once it meets all outstanding technical requirements.

Kind regards,

Dimitris Voudouris

Academic Editor

PLOS ONE

Additional Editor Comments (optional):

Reviewers' comments:

Reviewer's Responses to Questions

**Comments to the Author**

Reviewer #2: All comments have been addressed

2. Is the manuscript technically sound, and do the data support the conclusions?

Reviewer #2: Yes

3. Has the statistical analysis been performed appropriately and rigorously?

Reviewer #2: Yes

4. Have the authors made all data underlying the findings in their manuscript fully available?

Reviewer #2: Yes

5. Is the manuscript presented in an intelligible fashion and written in standard English?

Reviewer #2: Yes

Reviewer #2: I don’t have further comments. The interface here is a bit silly in that in requires to insert at least 100 characters.

**Do you want your identity to be public for this peer review?** For information about this choice, including consent withdrawal, please see our Privacy Policy

Reviewer #2: No
